# TempoRL: Temporal Priors for Exploration in Off-Policy Reinforcement Learning

## Abstract

Effective exploration is a crucial challenge in deep reinforcement learning. Behavioral priors have been shown to tackle this problem successfully, at the expense of reduced generality and restricted transferability. We thus propose temporal priors as a non-Markovian generalization of behavioral priors for guiding exploration in reinforcement learning. Critically, we focus on state-independent temporal priors, which exploit the idea of temporal consistency and are generally applicable and capable of transferring across a wide range of tasks. We show how dynamically sampling actions from a probabilistic mixture of policy and temporal prior can accelerate off-policy reinforcement learning in unseen downstream tasks. We provide empirical evidence that our approach improves upon strong baselines in long-horizon continuous control tasks under sparse reward settings.

## 1 Introduction

Exploration is a fundamental issue in reinforcement learning (RL): in order for an agent to maximize its reward signal, it needs to adequately cover its state space and observe the outcome of its actions. This becomes increasingly harder when dealing with large, continuous state and action spaces, which includes many real world applications. There exists a large and fruitful body of research on exploration (Bellemare et al., 2016; Osband et al., 2016; Tang et al., 2017; Osband et al., 2018; Azizzadenesheli et al., 2018; Burda et al., 2018; Dabney et al., 2021; Ecoffet et al., 2021), however most general-purpose algorithms remain based on $\epsilon$-greedy exploration (Mnih et al., 2015) or entropy-regularized Gaussian policies (Haarnoja et al., 2018). In the absence of an informative reward signal, both methods rely on uniformly sampling actions from the action space, independently of the history of the agent. Unfortunately, in sparse reward settings, achieving positive returns by uncorrelated exploration becomes exponentially less likely as the horizon length increases (Dabney et al., 2021). Moreover, within this setting, the lack of temporal correlation can result in undesirable behaviors during exploration, such as reversing recent actions.

A promising approach to achieve efficient exploration is that of using a behavioral prior to guide the policy (Pertsch et al., 2020; Tirumala et al., 2020; Singh et al., 2021). Typically, this is learned from expert trajectories as a state-conditional action distribution. Behavioral priors are able to foster directed and correlated exploration (Singh et al., 2021), by assuming a strong similarity between the agent and expert tasks. However, an agent should ideally be able to produce efficient explorative behaviors even in unseen environments and unrelated tasks.

To overcome the shortcomings of behavioral priors, we propose *temporal priors*, a powerful non-Markovian generalization which is capable of guiding exploration in challenging settings. In particular, we put our attention on the family of state-independent temporal priors, which enable accelerating reinforcement learning and transferring knowledge to unseen tasks by focusing on temporal correlation between actions.

In our method, which we dub TEMporal Priors for exploration in Off-policy Reinforcement Learning (TempoRL), we introduce a temporal action prior $\bar{\pi}(a_t|s_t, H_t)$, where $H_t = (s_i, a_i)_0^{t-1}$ is the history of the agent. In particular, we find that the class of state-independent temporal priors $\bar{\pi}(a_t|(a_i)_0^{t-1})$ is sufficient for capturing desirable properties for exploration, such as directness and temporal correlation, and advantageous in situations where the prior carries no knowledge about the current state. Temporal priors can be trained offline from few *task-agnostic* expert trajectories (see Figure 1). Furthermore, we propose a principled manner of integrating priors into the Soft Actor

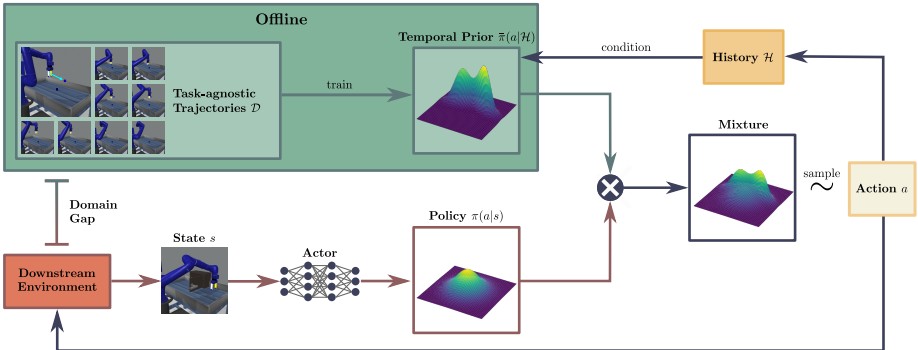

**Figure 1:** TempoRL: A temporal prior is trained on task-agnostic trajectories. Actions are then sampled from a dynamic mixture between a state-independent temporal prior and the policy in downstream learning of more complex tasks. Our method works with both vector-based and image-based state inputs.

Critic framework (Haarnoja et al., 2018) without breaking the Markovian assumption of the learning rule. In downstream learning of more complex tasks, our method samples actions from a dynamic mixture between the policy and the temporal prior. Moreover, the policy is regularized by directly maximizing the likelihood of sampling from the prior instead of the policy's entropy. Our approach is general and can be applied for arbitrarily complex priors.

In our experiments, we first compare the adequacy of behavioral priors and of different families of temporal priors for accelerating downstream RL. We then focus on state-independent temporal priors and verify their capability to produce correlated and directed behavior. We provide empirical evidence that our method can accelerate learning in long-horizon control tasks with sparse rewards. We demonstrate the effectiveness of our approach by comparing against state-of-the-art baselines.

Our contributions can be organized as follows:

1. We propose learnable non-Markovian action priors conditioned on the history of the agent. We show that sampling from these prior produces directed and correlated trajectories.

2. We introduce a novel manner of integrating temporal priors into the Soft Actor Critic framework (Haarnoja et al., 2018).

3. We show how state-independent temporal priors can be learned from few expert trajectories on simple tasks and used to improve exploration efficiency in new tasks, despite the presence of massive domain gaps [1] (e.g. from a simple reaching task to opening a window) and across entirely different settings (e.g., from non-visual to visual RL).

After discussing our method's novelty and related literature in Section 2, we introduce our setting in Section 3. The method is described in Section 4, while empirical evidence of its effectiveness is reported in Section 5. Finally, Section 6 contains a brief closing discussion of our work. We make our code available for research purposes [2].

## 2 RELATED WORK

**Temporally-Extended Exploration** Several works have attempted to directly address the inability of traditional methods, such as $\epsilon$-greedy or uniform action sampling (Lillicrap et al., 2016; Haarnoja et al., 2018), to produce correlated trajectories. An interesting study (Dabney et al., 2021) highlights this issue and shows how repeating random actions for multiple steps is sufficient to significantly accelerate Rainbow (Hessel et al., 2018) on Atari (Bellemare et al., 2013). Similarly, Amin et al. (2021) propose a non-learned policy inspired by the theory of freely-rotating chains in polymer physics to collect initial explorative trajectories in continuous control tasks. Both methods pinpoint a fundamental issue, but rely on scripted policies which are hand-crafted for a particular family of

---

[1]We borrow this term from domain adaptation literature to hint at the different nature of the environment used for collecting expert trajectories and of the environment the RL agent is deployed in. Informally, we consider the domain gap between two environments to be large if a policy trained in the first environment struggles to solve the second environment, or vice versa, even after additional training.

[2]https://sites.google.com/view/tempo-rl

environments. On the other hand, our method is learned from task-agnostic trajectories and does not require engineering an explorer, which can be unfeasible for complex tasks.

**Hierarchical Reinforcement Learning**   Another approach to tackle exploration-hard tasks is to rely on a hierarchical decomposition of the agent into different levels of *temporal* and *functional* abstraction (Parr & Russell, 1998; Dietterich, 2000; Sutton et al., 1999; Dayan & Hinton, 2000). For instance, the task can be decomposed into high level planning and a set of low-level policies, often referred to as *skills* (Konidaris & Barto, 2007; Eysenbach et al., 2018) or *options* (Sutton et al., 1999; Bacon et al., 2017). This approach effectively reduces the planning horizon and allows efficient solving of complex tasks from scratch (Bacon et al., 2017; Vezhnevets et al., 2017; Nachum et al., 2018; Levy et al., 2019; Christen et al., 2021). Low-level policies can be trained without supervision to achieve correlated and directed behaviors (Eysenbach et al., 2018), however, the issue of temporal correlation is merely relocated in the hierarchy, as the high-level planner is not encouraged to produce correlated sequences of skills. Incidentally, our method is not designed to achieve temporal abstraction, but can be interpreted in a hierarchical framework (Schäfer et al., 2021) in which a high-level criterion (the mixing function) governs a probabilistic choice between an explorer (temporal prior) and an exploiter (policy).

**Behavioral Priors**   Behavioral priors are generally represented by state-conditional action distributions modeling strategies for the current state of the environment. Such priors can be learned jointly with the policy in the context of KL-regularized RL (Tirumala et al., 2019; 2020), which in some cases restricts the information available to the prior (Galashov et al., 2019). A second approach consists in learning behavioral priors from expert policies on related tasks. This is the case for several works (Peng et al., 2019; Pertsch et al., 2020; 2021; Ajay et al., 2021) which adopt a Gaussian behavioral prior in a latent skill-space. In particular, Pertsch et al. (2020) report that a prior is crucial to guiding a high-level actor in an HRL framework. An important contribution to the field is made by PARROT (Singh et al., 2021), which focuses on a visual setup and introduces a flow-based transformation of the action space to allow arbitrarily complex prior distributions. We extend this idea to prior action distributions that are not only conditioned on the current state or a part thereof, but rather on the history of the agent, and are therefore non-Markovian. Moreover, we propose a novel and more flexible way of integrating the prior distribution into the learning algorithm. Most importantly, we overcome the reliance of the last two methods on a tight domain gap between tasks for training the prior and the agent.

## 3   BACKGROUND

### 3.1   SETTING

Reinforcement learning (RL) is the problem that an agent faces when learning to interact with a dynamic environment. Albeit with a slightly different definition, we formalize the environment as a goal-conditioned Markov Decision Process (gc-MDP) (Nasiriany et al., 2019), that consists of a 6-tuple $(\mathcal{S}, \mathcal{A}, \mathcal{G}, \mathcal{R}, \mathcal{T}, \gamma)$, where $\mathcal{S}$ is the state space, $\mathcal{A}$ is the action space, $\mathcal{G} \subseteq \mathcal{S}$ is the goal space, $\mathcal{R} : \mathcal{S} \times \mathcal{G} \to \mathbb{R}$ is a scalar reward function, $\mathcal{T} : \mathcal{S} \times \mathcal{A} \to \Pi(\mathcal{S})$ a probabilistic transition function that maps state-action pairs to distributions over $\mathcal{S}$ and, finally, $\gamma$ is a discount factor. Assuming goals to be drawn from a distribution $p_{\mathcal{G}}$, the objective of an RL agent can then be expressed over a time horizon $T$ as finding a probabilistic policy $\pi^{\star} = \arg\max_{\pi} \mathbb{E}_{g \sim p_{\mathcal{G}}} \sum_{t=0}^{t=T} \gamma^t \mathcal{R}(s_t, g)$, with $s_t \sim \mathcal{T}(s_{t-1}, a_{t-1})$ and $a_{t-1} \sim \pi(s_{t-1}, g)$. In order to simplify notation, from this point on, we will implicitly include the goal into the state at each time step: $s_t \leftarrow (s_t, g)$.

We focus on long-horizon control problems with continuous state and action spaces and sparse rewards, i.e., non-zero only after task completion. Although our method can be generally applied to stochastic off-policy RL methods, we build upon Soft Actor Critic (Haarnoja et al., 2018) with Hindsight Experience Replay (Andrychowicz et al., 2017), due to their wide adoption in these settings.

Finally, in contrast with several behavioral prior approaches (Galashov et al., 2019; Pertsch et al., 2020; Singh et al., 2021), we adopt a more general and challenging setting. First, we do not assume that prior information on the structure of the state space is available. Second, while we also assume access to a collection of expert trajectories $D = \{(s_0^i, a_0^i, s_1^i, a_1^i, \dots, s_N^i, a_N^i)\}_{i=0}^{L}$, we do not require high quality trajectories collected on the exact environment and task. From this point on, we refer

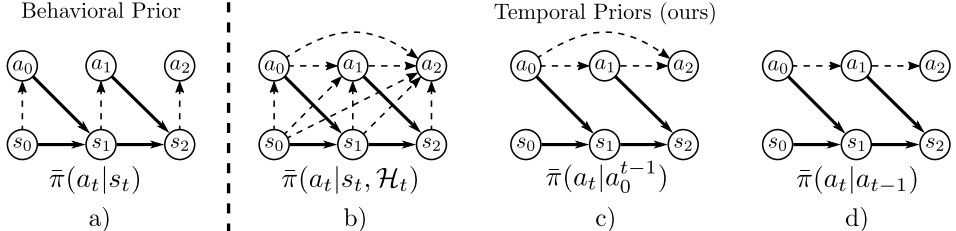

**Figure 2:** Graphical models representing different priors: from left to right, a behavioral prior, a general temporal prior, a state-independent temporal prior and a single-step state-independent temporal prior. Solid arrows represent the environment's transition function $\mathcal{T}$, while dashed arrow indicate conditional modeling.

to the environment used for collecting data as the *training environment*, and to the environment in which the RL agent is deployed as the *downstream environment*.

## 3.2 BEHAVIORAL PRIORS

A behavioral prior $\bar{\pi}(a|s)$ (Pertsch et al., 2020; Singh et al., 2021) is a state-conditional probability distribution over the action space (cf. Figure 2a). Behavioral priors can be trained to assign high probability to *useful* actions with respect to the current state, and hence be used to accelerate RL. A behavioral prior can only guide the policy effectively as long as prior information on the structure of the state space is available (Galashov et al., 2019), or the prior has been trained on data collected on a closely related task and environment (Pertsch et al., 2020; Singh et al., 2021).

In our settings, the structure of observations is unknown and expert trajectories $D$ may be collected on unrelated tasks. This means that the distribution of training states might not match the distribution of states produced by downstream environments: behavioral priors will then be evaluated on out-of-distribution samples and their performance will degrade drastically, as shown in Section 5.1.

## 4 METHOD

Our method relies on the integration of a learned temporal action prior into an off-policy RL algorithm. Thus, we first define and discuss the class of temporal priors of interest, and later describe how they can be integrated into existing off-policy algorithms.

### 4.1 TEMPORAL PRIORS

Within the settings outlined in Section 3.1, it is still possible to extract and transfer knowledge from the expert dataset $D$ to the agent, despite a significant domain gap. Namely, we can model the temporal correlation of the expert trajectories and use it to speed up downstream learning.

In order to recover this information, we thus propose to learn a *temporal prior*. A temporal prior is a non-Markovian action prior, representing a probability distribution over the action space that is not conditioned on the current state alone, but also on the past history of the agent: $\bar{\pi}(a_t|s_t, H_t)$ (see Figure 2b), where $H_t = (s_0, a_0, \ldots, s_{t-1}, a_{t-1})$. Temporal priors are a powerful generalization of behavioral priors, which directly address temporal correlation.

In the challenging settings we describe in Section 3.1, conditioning on the current state is not just insufficient, but might also be counterproductive, as a state-conditional action prior would receive out-of-distribution samples as inputs. Hence, we focus on the class of *state-independent temporal priors*, which drop their dependence on the environment's state. This is indeed a viable strategy in a hard-exploration setting, when no prior information is available on the state space and no reward is observed: in this case, no information can be extracted from the state in any case. The most general definition of a state-independent temporal prior is a probability density $\bar{\pi}(a_t|a_0^{t-1})$ (see Figure 2c). Our empirical evidence suggests that its simplest form, i.e., $\bar{\pi}(a_t|a_{t-1})$ (see Figure 2d) is surprisingly competitive with variants conditioned on multiple past actions (cf. Section 5.1) and is therefore sufficient to capture complex temporal relations.

Independently from the conditioning variables, temporal priors can conveniently be modeled as parametric or non-parametric conditional generative models and learned through empirical risk minimization. For the purpose of this paper, we choose to use the conditional variant of the Real Non Value Preserving Flow (Dinh et al., 2017; Ardizzone et al., 2019), which has been successfully applied in similar settings before (Singh et al., 2021) and is well suited for Euclidean action spaces. Our integration in the SAC framework allows arbitrarily complex prior distributions, which Real NVP Flows are in principle able to capture.

In the context of NVP Flows, training samples are actions $a \in \mathcal{A}$, paired with conditioning variables $(s, \mathcal{H})$, thus the learned mapping is $a = f_\theta(z; (s, \mathcal{H}))$, with $z \sim \mathcal{N}$. Since $f_\theta$ is invertible, we analytically compute the likelihood of a single training pair $(a, (s, \mathcal{H}))$ and maximize its expected value through standard gradient-based optimization techniques. An empirical justification of this choice is found in Appendix F.4, while implementation details are reported in Appendix D. For a complete introduction to Real NVP Flows, we refer the reader to Dinh et al. (2017).

One final concern regards the nature of the data for training the temporal prior. The main requirements for the training data are two: (1) the environment in which the data is collected needs to share the same action space of the downstream environment and (2) the training trajectories should display the desired qualities of correlation and directness. In general, we adopt task-agnostic expert trajectories generated by achieving simple random goals. Such simple trajectories can be learned from scratch using standard RL or, as we do in practice, produced by a scripted policy. We remark that, in contrast with existing approaches (Pertsch et al., 2020; Singh et al., 2021), this framework poses very weak requirements on the similarity between the environments used for data collection and the target environments. As we show in Section 5.3, this allows our method to bridge the gap between fundamentally different environments and settings, such as transferring from a simple reaching task with access to the true state of the system to a door-closing task in a visual RL setting.

## 4.2 SOFT ACTOR CRITIC WITH TEMPORL

The main challenge introduced by temporal priors stems from their non-Markovianity, which renders existing integrations of priors in RL unsuitable. Existing methods for accelerating RL through behavioral priors can only handle state-conditional distributions, which are modelled as Gaussians in most cases. For this reason, we introduce a novel method for the integration of an action prior in an off-policy RL framework. Our method is suitable for both behavioral and temporal priors, independently of their conditioning variables. The key strategy revolves around sampling actions from a mixture between the policy and a prior distribution, dynamically weighted through uncertainty estimation. We demonstrate it as an integration into the Soft Actor Critic framework.

---

**Algorithm 1** SAC with TempoRL

1: Train temporal prior $\bar{\pi}(a_t|s_t, H_t)$
2: Initialize history $H_0 = \emptyset$
3: Initialize policy and Q-parameters $\theta, \phi$
4: **for** each iteration **do**
5:     **for** each environment step **do**
6:         $\lambda_t = \Lambda(\mathcal{H}(\pi(\cdot|s_t)))$
7:         $a_t \sim (1 - \lambda_t)\pi(a_t|s_t) + \lambda_t \bar{\pi}(a_t|s_t, H_t)$
8:         $s_{t+1} \sim \mathcal{T}(s_t, a_t)$
9:         $D = D \cup (s_t, a_t, r(s_t, a_t), s_{t+1})$
10:        $H_{t+1} = H_t \cup (s_t, a_t)$
11:     **end for**
12:     **for** each gradient step **do**
13:         Update $\theta, \phi$
14:     **end for**
15: **end for**

---

SAC's objective is designed to pursue large rewards while maximizing the entropy of its policy. When prior knowledge on the structure of the environment or task is available, simply sampling actions from a maximal entropy policy $\pi$ may not be optimal. On the other hand, blindly sampling from a behavioral or temporal prior $\bar{\pi}$ prevents exploitation of reward signals as well as any behavior which is not encoded in the prior. Ideally, it is desirable to control the degree to which actions are sampled from the prior. We propose to achieve this in a natural way by sampling actions from a mixture between the policy $\pi$ and the prior $\bar{\pi}$:

$$a_t \sim (1 - \lambda_t)\pi(\cdot|s_t) + \lambda_t \bar{\pi}(\cdot|s_t, H_t) \quad \text{with } 0 \leq \lambda_t \leq 1, \tag{1}$$

where the mixing parameter $\lambda_t$ is computed dynamically at each step.

In principle, the current policy $\pi$ should be centered upon actions that maximize returns, while $\bar{\pi}$'s sole purpose is to suggest suitable actions according to behavioral or temporal knowledge. When observing a state $s_t$, the agent should then sample directly from its policy $\pi$ in case a path from such

state to the goal is known, and sample from $\bar{\pi}$ in case the state is unknown and further exploration is needed. For this reason, the mixing weight $\lambda_t$ should ideally estimate the probability of failing to reach the goal while only sampling from the policy $\pi$.

We therefore propose to compute the mixing weight directly as a function of the policy's entropy $\mathcal{H}(\pi(\cdot|s_t))$, which intuitively quantifies the agent's confidence in its plans. Since $\mathcal{H}(\pi(\cdot|s_t))$ cannot always be computed in closed form, we can estimate it via Monte Carlo sampling. While the number of samples can control the variance of the estimator, we simply use the current policy sample:

$$\mathcal{H}(\pi(\cdot|s_t)) = \mathop{\mathbb{E}}_{a_t \sim \pi}[-\log \pi(a_t|s_t)] \approx -\log \pi(a_t|s_t) \quad \text{with } a_t \sim \pi(\cdot|s_t). \tag{2}$$

The mixing weight can then be computed at each step as:

$$\lambda_t = \Lambda\big(\mathcal{H}(\pi(\cdot|s_t))\big) \approx \Lambda\big(-\log \pi(a_t|s_t)\big) \quad \text{with } a_t \sim \pi(\cdot|s_t), \tag{3}$$

where $\Lambda(\cdot)$ is a monotonically increasing mixing function bounded to the range $[0, 1]$.

We further incorporate this novel action sampling scheme by reformulating the objective to directly encourage sampling from the prior $\bar{\pi}$:

$$\pi^\star = \arg\max_\pi \mathop{\mathbb{E}}_{\tau \sim \pi}\left[\sum_{t=0}^{\infty} \gamma^t\bigg(\mathcal{R}(s_t, a_t) + \alpha\Lambda\big(\mathcal{H}(\pi(\cdot|s_t)))\big)\bigg)\right]. \tag{4}$$

Through straightforward derivations (see Appendix A), one can retrieve a modified objective for training the policy $\pi$ and a Q-function estimator $Q^\pi$ in the Soft Actor Critic framework. Given a distribution $\mathcal{D}$ of observed states and actions, the loss functions can be defined as:

$$J_\pi = -\mathop{\mathbb{E}}_{s \sim \mathcal{D}}\left[Q_\phi^\pi(s, a) + \alpha\Lambda(-\log \pi_\theta(a|s))\right] \quad \text{with } a \sim \pi_\theta(\cdot|s), \tag{5}$$

$$J_Q = \mathop{\mathbb{E}}_{(s,a) \sim \mathcal{D}}\left[\big(Q_\phi^\pi(s, a) - y_t(s, a)\big)^2\right], \tag{6}$$

where the target for the Q-value is computed as

$$y_t(s, a) = \mathcal{R}(s, a) + \gamma\bigg(Q_\phi^\pi(s', a') + \alpha\Lambda\big(-\log \pi_\theta(a'|s')\big)\bigg) \quad \text{with } s' \sim \mathcal{T}(\cdot|s), a' \sim \pi_\theta(\cdot|s'). \tag{7}$$

We note that minimizing Equations 5 and 6 introduces a bias with respect to the objective in Equation 4 for non-linear mixing functions due to Monte Carlo estimation of entropy. The two objectives can be empirically estimated and minimized through standard procedures, as reported in Haarnoja et al. (2018) and in Appendix A.

Algorithm 1 summarizes (in blue) the modifications to be applied in order to integrate our prior into the SAC framework. Namely, actions are sampled from a mixture (line 7) weighted according to the output of a mixing function (line 6). Finally, the history of the agent needs to be initialized (line 1) and updated at each step (line 10). Update rules for $\theta$ and $\phi$ are modified and computed from Equations 5 and 6.

We finally note that the modified learning objective remains aligned with the original formulation. As a consequence, while our method is superior on more complex tasks (cf. Section 5.3), leaving SAC's objective unchanged can perform on-par on some tasks, with reduced sensitivity to hyperparameter tuning. We refer the reader to Appendix B for more details.

**Mixing Function** The output of the mixing function $\Lambda$ should be constrained to the range $[0, 1]$ and monotonically increasing with respect to its input. Intuitively, if the entropy of the policy increases (i.e. in the absence of a strong reward signal), the mixing weight should also increase, as sampling from the prior becomes more desirable. A natural choice is to simply apply a sigmoid function after scaling and translating the entropy via parameters $\beta_t, \beta_s$. These parameters are estimated empirically and kept fixed across all experiments. The resulting function for computing the mixing weights is:

$$\Lambda\big(\mathcal{H}(\pi(\cdot|s_t))\big) = \sigma\big(\beta_s \mathcal{H}(\pi(\cdot|s_t)) - \beta_t\big). \tag{8}$$

**Figure 3:** Overview of environments used in our experimental validation.

# 5 EXPERIMENTS

We evaluate our method in a series of experiments to empirically validate our contributions. First, in Section 5.1, we compare the effectiveness of behavioral and temporal priors to justify our choice of state-independent conditioning. In Section 5.2, we verify that sampling from our temporal prior produces correlated and state-covering behavior, without the need to hand-craft an exploration policy. Next, we show how our method can improve exploration efficiency in unseen long-horizon tasks by comparing against various baselines in state-based RL (Section 5.3). Furthermore, we demonstrate that our prior enables transfer to different settings, i.e., from non-visual to visual state space, whilst retaining the aforementioned benefits in exploration. An ablation of the generative model is provided in Appendix F.4.

**Baselines**   We now present the baselines. Implementation details are provided in Appendix D.3.

- **SAC**: vanilla Soft Actor Critic (Haarnoja et al., 2018)
- **SAC+AR($n$)**: SAC with a non-learning based prior that repeats an action $n$ times to enforce more directed behavior. We choose $n = 2$ for our experiments.
- **SAC+BC**: SAC with warm-started policy through behavior cloning.
- **SAC-PolyRL**: SAC with locally self-avoiding walks (Amin et al., 2021).
- **PARROT-state**: flow-based behavioral prior enforced through a transformation of the action space (Singh et al., 2021). We benchmark a state-based variant in non-visual settings.

**Environments**   We evaluate our method on two types of domains, namely robotic manipulation and maze navigation. Specifically, we make use of a subset of robot manipulation tasks from the publicly available `meta-world` suite (Yu et al., 2020) and adapt the `point-maze` implementation from Pitis et al. (2020). More details can be found in Appendix C, while visual examples of the environments are provided in Figure 3.

Our temporal priors, as well as prior-based baselines, are learned in relatively simple *training environments* that make very weak assumptions about the environment and task structure. To this end, we train priors for robot manipulation in a reaching task (`reach`) and for maze navigation in an empty environment (`room-maze`), where the task is completed upon reaching a goal that is sampled uniformly from the whole environment space.

RL agents are then trained and evaluated on a wider range of *downstream environments* and tasks. For analyzing how learned priors can improve exploration efficiency, we deploy an RL agent on both the environment used for training the priors and a set of unseen, more difficult test tasks. The latter consist of manipulating objects, such as opening a window, or navigating in more complex mazes.

## 5.1 CONDITIONING VARIABLES FOR BEHAVIORAL AND TEMPORAL PRIORS

The goal of this section is to empirically show how the effectiveness of an action prior depends on conditioning variables and on the domain gap between the training and the downstream environment. For this purpose, we train several variants of flow-based action priors on the robotic reaching task (`reach`) and train TempoRL from scratch on the same environment, as well as on a different one (`window-open`). In particular, we compare temporal priors conditioned on action sequences of different lengths (1, 2, 5, 10), temporal priors conditioned on the previous state-action pair and a behavioral prior (conditioned on the state alone).

The results are provided in Figure 4. We observe that both types of priors are capable of guiding downstream RL as long as the downstream environment matches the training environment. However, we find that including the state in the conditioning variables can jeopardize the ability to trans-

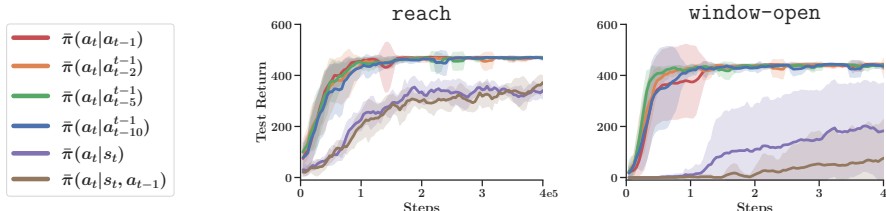

**Figure 4:** Comparison of downstream performance of behavioral and temporal priors when conditioned on actions, states and combinations thereof.

fer knowledge to a different task, such as `window-open`. We hypothesize that this is due to a mismatch between the states used for training the prior and those returned by the downstream environment. For this reason, behavioral priors $\bar{\pi}(a_t|s_t)$ fail on the unseen task. Even when the prior is conditioned on state-action pairs $\bar{\pi}(a_t|s_t, a_{t-1})$, it learns to rely on its state input to model the action distribution, and therefore fails to transfer. Furthermore, we observe that state-conditional priors suffer when training in the same environment that was used for data collection, which we argue is due to the limited variance in sampled trajectories at training time (see Appendix D.4).

On the other hand, state-independent temporal priors prove to be a capable alternative across both settings and are able to transfer knowledge to unseen tasks. While conditioning on longer action sequences can improve performance, we note that single-action-conditional models $\bar{\pi}(a_t|a_{t-1})$ are sufficient for capturing complex temporal dependencies within our settings. Hence, they will be the focus of the remaining experiments.

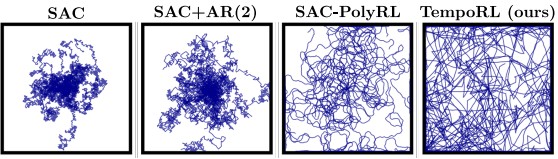

**Figure 5:** A qualitative comparison of sampled exploration trajectories in a 2D room. Our method achieves directed behavior while covering most of the state space. SAC and SAC+AR(2) fail to cover the full state space, while SAC-PolyRL fails to reach distant areas consistently.

**Table 1:** State coverage metrics for our method and baselines. TempoRL's trajectories are locally directed and cover the state space well in both environments.

|  |  | $U_g^2$ | % Coverage |
|---|---|---|---|
| reach | SAC | 0.006±0.001 | 0.137±0.01 |
|  | SAC+AR(2) | 0.010±0.002 | 0.199±0.02 |
|  | SAC-PolyRL | 0.025±0.001 | 0.272±0.01 |
|  | TempoRL (ours) | **0.053±0.009** | **0.357±0.02** |
| maze | SAC | 0.005±0.001 | 0.333±0.02 |
|  | SAC+AR(2) | 0.008±0.001 | 0.493± 0.08 |
|  | SAC-PolyRL | 0.026±0.002 | 0.880±0.04 |
|  | TempoRL (ours) | **0.054±0.008** | **0.963±0.04** |

## 5.2 CORRELATION AND STATE COVERAGE

In this experiment, we show how a one-step state-independent temporal prior $\bar{\pi}(a_t|a_{t-1})$ produces correlated and directed behavior, which leads to a more complete coverage of the state space during exploration. To this end, we sample 20 random trajectories of 500 steps each with our method and several baselines in the `room-maze` and `reach` environment.

As shown in Figure 5, our temporal prior produces directed behavior which covers most of the state space. As expected, uniform sampling (SAC) and action repeat (SAC+AR(2)) fail to reach the boundaries of the environment. SAC-PolyRL is capable of producing correlated and directed behaviors, but only after careful hyperparameter tuning. Equivalent behavior can be observed in the robot manipulation environment (cf. Appendix F.7). This qualitative assessment is verified quantitatively in the evaluation presented in Table 1. We report state space coverage and radius of gyration squared (Amin et al., 2021) (see Appendix D.1 for details). Evidently, our method outperforms the presented baselines on both metrics.

## 5.3 TRANSFER LEARNING

Our main results are obtained by comparing our method against several baselines in downstream learning tasks with a vectorized state space, as presented in Figure 6. As expected, we observe that the performance of behavioral priors depends on the similarity of the downstream task with the expert dataset. This is the case for PARROT-state, which solves `reach` and `room-maze` easily, as

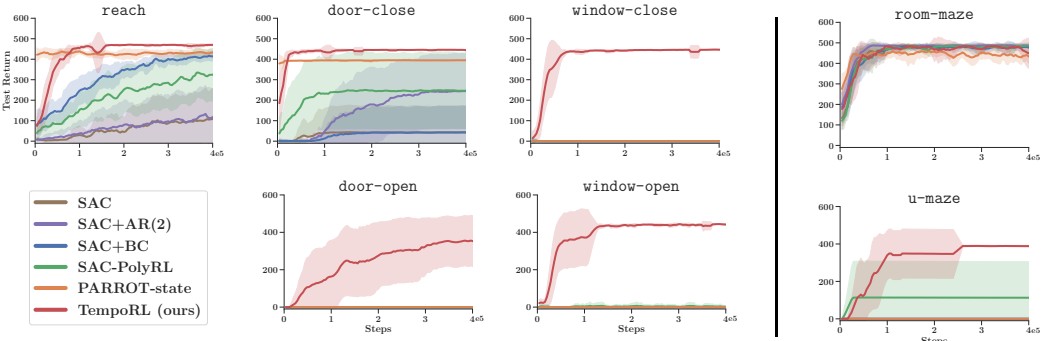

**Figure 6:** Accelerating downstream RL. While other methods are mostly competitive on training tasks (`reach` and `room-maze`), TempoRL is able to accelerate RL for unseen tasks (`door-close`, `door-open`, `window-close`, `window-open`, `u-maze`).

its behavioral prior already represents a strong policy. PARROT-state also offers good performance in `door-close`, as the act of reaching for the goal, which is the final position of the door, is sufficient to swing it closed. On tasks which are significantly different from the training task, namely `window-open`, `window-close` and `u-maze`, PARROT-state is unable to guide the policy, as it receives out-of-distribution states (cfr. Appendix C). On the other hand, TempoRL is capable of transferring to unseen tasks, while rapidly catching up with PARROT-state in the training tasks.

Other baselines are in general less effective across the benchmarks: Vanilla SAC only makes progress in the reaching task, due to the presence of easily reachable goals that can be achieved even with weak exploration. Enabling action repeat (SAC+AR(2)) can effectively speed up exploration, but only on tasks that are slightly out of reach when the heuristic is not enabled. SAC-PolyRL is able to produce good explorative trajectories through its hand-crafted policy, but its performance is strongly dependent on the task. As previously reported by Singh et al. (2021), initializing SAC through behavioral cloning (SAC+BC) can help guide exploration, but it fails to generalize across tasks and is regularly outperformed by stronger methods.

**Visual RL** We finally demonstrate the benefits of temporal priors in more complex settings. In particular, state-independent temporal priors allow the state space of the downstream environment to be defined arbitrarily. Hence, they also allow transfer to the visual RL setting, which avoids reliance on a low-dimensional vectorized state space and is purely based on RGB observations. To this end, we compare TempoRL with Vanilla SAC in Figure 7 and with PARROT in its original, visual setup, i.e., by conditioning its behavioral prior on images. We report that our findings hold in visual settings: temporal priors are capable of generalizing to unseen tasks, even when transferred to fundamentally different state spaces. More results are presented in Appendix F.8.

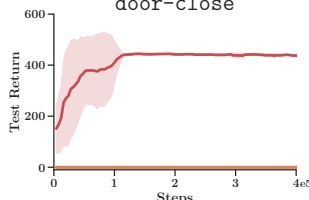

**Figure 7:** Transfer of a one-step state-independent temporal prior to Visual RL. TempoRL (red) compared with SAC (brown) and PARROT (orange).

## 6 CONCLUSION

In this paper, we propose a method for improved exploration efficiency in off-policy reinforcement learning. In particular, we introduce a non-Markovian, flow-based temporal prior and show how it can be integrated into an off-policy reinforcement learning framework. Crucially, we argue that state-conditioned priors struggle with transferring knowledge across domain gaps and provide empirical evidence on how a state-independent temporal prior can accelerate learning in unseen long-horizon control tasks with sparse rewards. As our method shows promising results, there are exciting directions for future work. State-independent temporal priors demonstrated their usefulness in unseen tasks, to which behavioral priors often cannot extrapolate. On the other hand, state-conditioned prior can directly transfer knowledge in the absence of a domain gap. Since both families of methods remain strong in complementary settings, we hope to explore the direction of a flexible prior, capable of both general and task-specific exploration.

## 7 ETHICS STATEMENT

Our main contribution revolves on accelerating and enabling reinforcement learning in environments with a strong exploration component. As a consequence, we believe that concerns with respect to our method are for the most part aligned with general RL research. For instance, improved sample efficiency could on one hand accelerate the process of automation, which might have a negative impact on societal equality, and on the other hand democratize access to powerful RL methods by lowering the amount of required resources. Due to the general nature of our method, we believe that we do not introduce fundamentally new risks.

## 8 REPRODUCIBILITY STATEMENT

We aim to achieve full reproducibility in our experiments. In practice, we extensively describe implementation details in Appendix D and, most importantly, we make our codebase public for research purposes (see URL in Footnote 2).

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

# A    OBJECTIVE DERIVATION

The purpose of this section is that of describing how learning objectives for the policy and Q-estimators can be derived from the TempoRL objective:

$$\mathbb{E}_{\tau \sim \pi} \left[ \sum_{t=0}^{\infty} \gamma^t \Big( R(s_t, a_t) + \alpha \Lambda \big( \mathcal{H}(\pi(\cdot | s_t)) \big) \Big) \right]. \tag{9}$$

The steps of these derivations follow those reported in Haarnoja et al. (2018), as our method is designed to fit into their framework. First, let us formally introduce the Q-function for a generic policy $\pi$:

$$Q^\pi(s, a) = \mathbb{E}_{\tau \sim \pi} \left[ \sum_{t=0}^{\infty} \gamma^t R(s_t, a_t) + \alpha \sum_{t=1}^{\infty} \gamma^t \Lambda \big( \mathcal{H}(\pi(\cdot, s_t)) \big) | s_0 = s, a_0 = a \right]. \tag{10}$$

We can then formulate the Bellman Equation and explicitly unravel the entropy term:

$$\begin{aligned} Q^\pi(s, a) &= \mathbb{E}_{s' \sim \mathcal{T}, a' \sim \pi} \left[ R(s, a) + \gamma(Q^\pi(s', a') + \alpha \Lambda \big( \mathcal{H}(\pi(\cdot, s')) \big)) \right] \\ &= \mathbb{E}_{s' \sim \mathcal{T}, a' \sim \pi} \left[ R(s, a) + \gamma(Q^\pi(s', a') + \alpha \Lambda \big( \mathbb{E}_{a'' \sim \pi} - \log \pi(a'' | s') \big)) \right]. \end{aligned} \tag{11}$$

The right-hand side of Equation 11 can be estimated via Monte Carlo sampling and used as a target, as shown in Equation 7. Q-estimates can be trained by minimizing a standard MSE error, resulting in the Q-loss in Equation 6.

In practice, as is done for SAC, two separate parameterized Q-function estimators $(Q_{\phi_i})_{i=1,2}$ are used to prevent overestimating Q-values; the policy is also parameterized as $\pi_\theta$. Additionally, we also adopt target networks $(Q_{\phi_{\text{target}, i}})_{i=1,2}$, updated via Polyak averaging. As a result, when sampling a batch $B$ from a replay buffer, the empirical estimates for the Q-losses are:

$$\hat{J}_{Q_{\phi_i}} = \frac{1}{|B|} \sum_{(s, a, r, s', d) \in B} \left( Q_{\phi_i}(s, a) - \hat{y}_t(s', r, d) \right)^2, \tag{12}$$

where the target for the Q-value is computed as

$$\hat{y}_t(s', r, d) = r + \gamma(1 - d) \left( \min_{i=1,2} Q_{\phi_{\text{target}, i}}(s', a') + \alpha \Lambda \big( -\log \pi_\theta(a' | s') \big) \right) \quad \text{with } a' \sim \pi_\theta(\cdot | s'). \tag{13}$$

and $r, d$ stand for the reward and done signal, respectively.

On the other hand, the policy $\pi_\theta$ can be trained to minimize the KL-divergence with the soft-max of the Q-function (Haarnoja et al., 2019), or alternatively to maximize the value function (Haarnoja et al., 2018). Our method relies on the second option and trains $\pi$ to maximize its value function $V^\pi(s)$, which can be formulated as follows:

$$V^\pi(s) = \mathbb{E}_{a \sim \pi} \left[ Q^\pi(s, a) + \alpha \Lambda (\mathcal{H}(\pi(\cdot | s))) \right]. \tag{14}$$

Optimizing over a distribution of states $s \sim \mathcal{D}$ results in the biased objective in Equation 6. Computing the expectation over actions can then be circumvented by using the reparameterization trick,

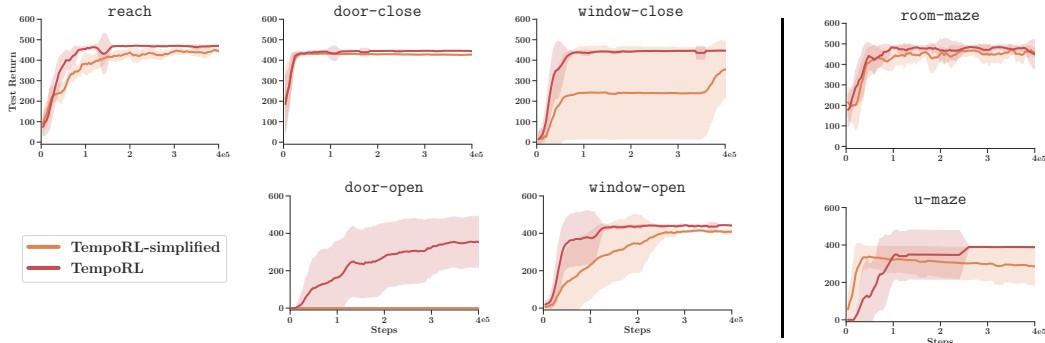

**Figure 8:** Comparison between the full method and its simplified version. The simplified version does not modify SAC's update rules.

which enables expressing actions as $a_\theta(\xi, s)$, where $\xi \sim \mathcal{N}$ is sampled from a standard Gaussian. In practice, since the entropy term can once again be estimated by Monte Carlo sampling, this finally translates into the empirical loss estimate:

$$\hat{J}_{\pi_\theta} = -\frac{1}{|B|} \sum_{(s)\in B} \left( \min_{i=1,2} Q_{\phi_i}(s, a_\theta(s, \xi)) + \alpha\Lambda\big( -\log \pi_\theta(a_\theta(s, \xi)|s)\big) \right), \tag{15}$$

which can be minimized via standard first-order optimization techniques.

## B  SIMPLIFIED ALGORITHM

We note that a policy $\pi^\star$ maximizing Objective 4 is not necessarily a maximizer for the original objective in SAC Haarnoja et al. (2018), which is:

$$\mathbb{E}_{\tau\sim\pi} \left[ \sum_{t=0}^{\infty} \gamma^t \bigg( R(s_t, a_t) + \alpha\mathcal{H}(\pi(\cdot|s_t)) \bigg) \right]. \tag{16}$$

However, as we note in Section 4, due to monotonicity of the mixing function $\Lambda(\cdot)$, the regularization term in the TempoRL objective and the regularization term in SAC are both maximized by a max-entropy policy $\pi^\star = \mathcal{U}(\mathcal{A})$. This observation suggests that modifying the learning objective might not be essential to our method. We thus set out to validate this hypothesis empirically, by removing all modifications to the objective or the learning rules of SAC. The resulting algorithm, which we refer to as *TempoRL-simplified*, was validated on the same settings of Section 5.3 in Figure 8. In general, we find that the simplified version is mostly on par with the full method in terms of downstream performance on the easier tasks (e.g. door-close), but it is not able to learn the more complex tasks (e.g. door-open). Interestingly, we find the simplified version to be more robust with respect to changes in hyperparameters, and we would suggest it as a practical alternative to the full method on easier tasks.

## C  ENVIRONMENT DETAILS

### C.1  ROBOT MANIPULATION

We use the meta-world suite (Yu et al., 2020) for robot manipulation experiments. It consists of a simulated 7 DoF Sawyer arm, implemented in the MuJoCo physics engine (Todorov et al., 2012). By default, states are represented as 36-dimensional vectors across all tasks. The state contains the 3D location and aperture of the gripper, the 3D location and quaternion of one object (e.g. door or window), measured for the current and previous time step. Goals are represented as the desired

3D location of the end effector or object, according to the task. Actions are 4-dimensional vectors containing a 3D movement and a 1D control over the aperture of the effector.

We additionally render 64x64 RGB images as observations for the visual setup in Section 5.3, using the camera angle `corner` (as can be seen in Figure 3). Instead of the originally proposed dense reward, we adopt a sparse reward which is only non-zero upon task completion. For more details, we refer to the original implementation (Yu et al., 2020).

## C.2 MAZE NAVIGATION

Our `point-maze` environments were adapted from Pitis et al. (2020). The state consists of the 2D position of the agent, which can be actuated via a velocity-controller through 2D actions. The goal space matches the state spaces in dimensions and representation. The reward signal is 1 when the Euclidean distance to the goal is lesser than 1.2 units, else it is 0. We experiment with two different layouts (renderings can be found in Figure 3):

- `room-maze` is a large, 29×29 square room. The starting position is at the center and the goal is initialized randomly in one of the 4 corners at each reset. Trajectories for training the priors are obtained from this environment. Only for Figure 5, the size was increased to 81×81, to allow and better visualize long trajectories.

- `u-maze` is a larger u-shaped corridor with three parts of lengths 50, 3 and 50 respectively, assembled at 90° clockwise rotations. The structure of the maze can therefore be contained in a $50 \times 3$ rectangle. Starting and goal position are fixed and located at opposite ends of the maze.

For further details, we refer directly to our published codebase (URL in Footnote 2).

## D IMPLEMENTATION DETAILS

### D.1 METRICS

All metrics reported in this paper (e.g. cumulative returns per episode) are averaged over 7 random seeds (or 9 for Figure 6); mean and standard deviation are reported and plots are smoothed over 5 steps.

We rely on two metrics for measuring the quality of exploration trajectories in Section 5.2:

- **%Coverage** divides the reachable state space in $n$ cubic buckets ($n = 1000$ for `reach` and $n = 100$ for `room-maze`) and reports the ratio between the number of buckets visited by a set of 20 trajectories of length 500 and the total number of buckets.

- **Radius of Gyration Squared** measures the spread in visited states, averaged over all trajectories, and is adapted from Amin et al. (2021). Given a set $T$ of $n$ trajectories, the metric can be computed as:

$$U_g^2(T) = \frac{1}{\delta n} \sum_{\tau \in T} \frac{1}{|\tau| - 1} \sum_{s \in \tau} d^2(s, \bar{\tau}),$$

where a trajectory $\tau$ is modeled as a sequence of states $(s_i)_0^{|\tau|-1}$, $d^2(\cdot, \cdot)$ measures the Euclidean distance and $\bar{\tau} = \frac{1}{|\tau|} \sum_{s \in \tau} s$. We additionally normalize the metric by $\delta$, which measures the diagonal of the box containing reachable states.

### D.2 DATA COLLECTION

Although the training tasks are relatively simple and a data collection policy can in principle be trained from scratch, for simplicity we fill the expert dataset $\mathcal{D}$ with the trajectories of a scripted policy. Since training environments do not include obstacles and allow direct control over the agent position(in 2D for `room-maze` and in 3D for `reach`), scripted policies simply receive the 2D/3D positions of the agent and of its goal, and output a distance vector. This vector is then corrupted with isotropic Gaussian noise and scaled to fit within action limits. A simple implementation of scripted

policies is available as part of our codebase (see URL in Footnote 2). For each training environment (`reach` or `room-maze`), we collect 4000 trajectories of 500 steps each. Goals for the scripted policy are sampled uniformly from the reachable state space. On goal achievement, a new goal is sampled, once again uniformly. The same expert datasets are then used for training temporal priors, behavioral priors or behavioral cloning.

## D.3 BASELINES

**SAC** We build upon the implementation provided by SpinningUp (Achiam, 2018), which is reported to be roughly on-par with the best results achieved on MuJoCo Gym (Brockman et al., 2016). For simplicity, we do not use automatic entropy tuning and keep $\alpha$ constant during learning. Nonetheless, our method could easily tune $\alpha$ dynamically at the expense of increased complexity, as is described in Haarnoja et al. (2018). Moreover, we also introduce HER (Andrychowicz et al., 2017) as well as n-step returns (Hessel et al., 2018). We experimented with importance sampling for off-policy correction, but, similarly to what is reported by Hessel et al. (2018), we observed no empirical benefit. By default, we sum rewards over $n = 10$ steps before clipping the range to $[0, 1]$. All remaining hyperparameters are reported in Table 2. We anticipate that all baselines also rely on HER and n-step returns as just described.

**Table 2:** Hyperparameters for SAC.

| Hyperparameter | Value |
|---|---|
| Epochs | 125 |
| Steps Per Epoch | $4e3$ |
| Steps of Initial Exploration | $1e4$ |
| Steps Before Training | $1e3$ |
| Environment Steps per Iteration | 50 |
| $\gamma$ | 0.99 |
| Polyak Averaging Rate | 0.995 |
| Inverse of Reward Scale ($\alpha$) | 0.2 for `meta-world`, 0.02 for `point-maze` |
| Batch Size | 100 |
| Optimizer | Adam |
| $\beta_1$ | 0.9 |
| $\beta_2$ | 0.999 |
| Learning Rate | 0.001 |
| Hidden Units | 256 |
| Hidden Layers | 2 |
| Hindsight Replay Ratio | 4 |
| Replay Size | $5e5$ for vector-based RL, $2e5$ for image-based RL |

**SAC+AR(n)** This baseline shares all hyperparameters with SAC. The only difference lies in the fact that the policy is only sampled from every $n$-th step, while the previous action is repeated for the remaining steps. We use $n = 2$ in our experiments.

**SAC+BC** Behavioral cloning (BC) is performed on the entire dataset $\mathcal{D}$ for 10 epochs, by maximizing the log-likelihood of the Gaussian policy with respect to expert actions. The optimizer and batch size used for BC are the same as for downstream learning.

**SAC-PolyRL** SAC-PolyRL (Amin et al., 2021) replaces the initial uniform exploration phase of SAC with trajectories collected by a hand-crafted policy. While SAC's hyperparameters are unvaried, SAC-PolyRL specific parameters are tuned from those reported in various settings in the original paper. We use $\theta = 0.35$, $\sigma^2 = 0.017$ and $\beta = 0.01$.

**PARROT** An official implementation for PARROT (Singh et al., 2021) is not available at the time of writing. We reproduce their training routine and downstream application and adopt all hyperparameters reported in the original paper. Generative models are trained until convergence (100 epochs) using a batch size of 400 samples and Adam (Kingma & Ba, 2017) as an optimizer, with a learning rate of 0.0001, $\beta_1 = 0.9$, $\beta_2 = 0.999$ and a weight decay penalty of $1e - 6$.

In addition to the image-based version of PARROT used in Section 5.3, we introduce a version with a vectorized state space, dubbed PARROT-state, since the method is originally only applicable in visual settings. The only modification consists in replacing the image encoder with a 3-layer MLP with 256 neurons per layer and ReLU activations. In this setting, the input to the encoder is therefore vector-based (non-visual).

### D.4 TEMPORL

**Prior**   We model all families of temporal priors with conditional Real NVP Flows (Ardizzone et al., 2019), sharing the same architecture across all experiments. The invertible transformation $f_\theta$ is a composition of 6 coupling layers, each followed by a batch-normalization layer (Dinh et al., 2017). For each layer, a 3-layer MLP with 128 hidden units per layer is used to preprocess the conditioning input. Scale and transition networks are also implemented as a shared 3-layer MLP with 128 hidden units, whose output is split in two to provide scale and shift coefficients. All MLPs use ReLU activations.

We train our temporal priors following the same protocol used with behavioral priors (100 epochs, a batch size of 400 samples and Adam (Kingma & Ba, 2017), with a learning rate of 0.0001, $\beta_1 = 0.9$, $\beta_2 = 0.999$ and a weight decay penalty of $1e - 6$).

**Downstream Learning**   Our method introduces a pair of hyperparameters governing the mixing function $\Lambda(\cdot)$, which we tune on `window-open` and apply to all other tasks and environment: $\beta_t = -0.7$, $\beta_s = \frac{1}{0.75|\mathcal{A}|}$, where $\mathcal{A}$ is the action space. Due to the modifications to the objective, we also tune $\alpha$ to 0.2 for `point-maze` and 1.0 for `meta-world`. While SAC samples action uniformly for the first 10000 steps to encourage exploration, TempoRL has access to a temporal prior $\bar{\pi}$ and therefore, during this initial phase, we sample directly from it in `meta-world`. To overcome stability issues, TempoRL still samples initial actions uniformly and adopt a lower learning rate of 0.0001 in `point-maze`. Among these changes, those that could also be applied to the baselines were tested and found to not be beneficial to them. All remaining hyperparameters are shared with SAC (see Table 2).

**Test-time Policy**   At test time, instead of sampling an action from the mixture between prior $\bar{\pi}$ and policy $\pi$, the agent executes an action corresponding to the mean of the policy distribution. We note that this is consistent to what is done in SAC Haarnoja et al. (2018).

**Simplified Method**   The simplified version uses $\beta_t = 0.65$ and $\beta_s = \frac{40}{|\mathcal{A}|}$, while all remaining hyperparameters are shared with SAC. Similarly to the full method, it also samples initial actions from the prior.

**State-conditional Prior**   When using state-dependent priors $\bar{\pi}(a_t|s_t)$ and $\bar{\pi}(a_t|s_t, a_{t-1})$ to accelerate downstream learning in Section 5.1, trajectories at training time are immediately near-optimal and often distributed over an excessively small support, which hurts test-time performance. To mitigate this issue, when using state-dependent priors in our framework, we multiply the mixing weight $\lambda_t$ by a scheduling coefficient $\epsilon$, which we linearly increase from 0 to 1 over the duration of training. Intuitively, this leads to more diverse initial trajectories.

## E   LIMITATIONS

Section 5 empirically argues in favor of our method's ability to produce correlated trajectories and accelerate downstream learning in unseen tasks. While state-independent temporal priors are able to generalize across larger domain gaps with respect to baselines, they can only be as general as the data used for training. In other words, temporal priors are data-driven, and as such will only reconstruct behavior that appears in the offline dataset of trajectories they are trained on. For instance, when training on `reach` with random goals, temporal priors will not encode any grasping strategy. While such behavior can still be recovered by the policy $\pi$, there is no incentive to do so. As a result, successfully transferring to tasks that require additional strategies such as grasping (e.g. `pick-place`) remains an open challenge, as we show in Appendix F.1.

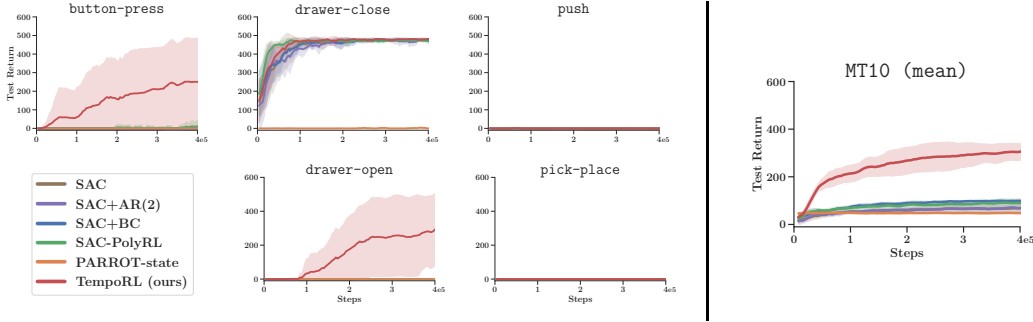

**Figure 9:** Left: performance on additional downstream tasks from MT10 (Yu et al., 2020). Right: mean performance over 9 out of 10 tasks in MT10 (`peg-insert-side` does not allow hindsight relabeling). Shaded area represents variation over seeds. We note that our training and evaluation protocol is presented in Section 5.3 and does not rely on a single, multi-task policy.

More generally, temporal priors can still suffer from domain gaps, especially when they are state-dependent and there is a strong correlation between the current state and the expert action during training. In this case, the prior model's output might rely on its state input, and therefore suffer from out-of-distribution samples during downstream learning. State-independent temporal priors can alleviate this issue. On the other hand, they might provide weaker guidance on tasks that are very similar to the training task since they are agnostic to the state. As shown in Figure 6, behavioral priors are in practice superior when the downstream environment can be solved by the policy used for collecting expert trajectories. For this reason, a strategy to combine benefits from behavioral and temporal priors would represent an interesting direction for the future.

One final concern regards entropy quantification. As we discuss in Section 4, mixing weights $\lambda_t$ should represent the confidence of the agent in its plans. For simplicity, we estimate this through the policy's entropy in the SAC framework. While we empirically found this choice to be effective, in certain settings (e.g. when multiple actions lead to large rewards), the policy's entropy might remain high even when the agent is capable of solving its environment. This might lead to overly sampling from the temporal prior, which is undesirable in this situation. However, in practice, this is a rare occurrence when dealing with long-horizon, sparse reward tasks, in which strong policies also need to be precise. We did not find this issue to be significant in the context of experimental validation.

## F  ADDITIONAL RESULTS

### F.1  ADDITIONAL TASKS FROM META-WORLD

Our selection of `meta-world` tasks is meant to keep the evaluation in Section 5 concise. We select the (arguably) easiest task for training (`reach`) and all tasks on two representative objects (door and window) as additional downstream tasks. We now report and discuss performance for the remaining tasks in MT10, adopting the settings used in Section 5.3,

Additional results remain consistent with our previous experiments. In particular, TempoRL performs well on two tasks that are out of reach for all baselines. All methods fail in environments that require grasping or precise manipulation. We note that this is an expected result of training a state-independent temporal prior on a task that does not require actuating the gripper. Please note that `pick-place` could not be included in this comparison, as it was not possible to extract its reward function for hindsight relabeling.

### F.2  ADDITIONAL TASKS FROM POINT-MAZE

We found our results in Section 5.3 to remain valid for different maze structures. In Figure 10 we visualize an additional task (a `complex-maze` of size $28 \times 28$) and we report results for TempoRL and all baselines.

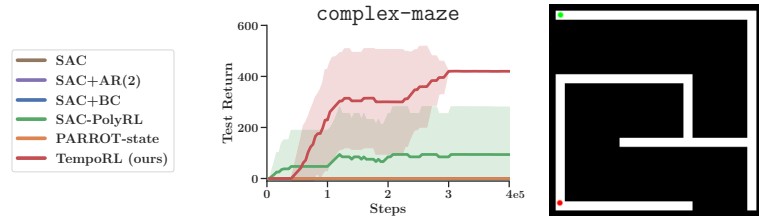

**Figure 10:** Performance on an additional maze structure (`complex-maze`). Settings are identical to those presented in Section 5.3.

## F.3 ADDITIONAL TASKS FROM GYM-MUJOCO

In this section, we apply our method in a substantially different setting, represented by the `swimmer-v2` environment from the gym-Mujoco suite Brockman et al. (2016). While an extensive evaluation on tasks from this suite is outside of the scope of this paper, we believe that this represents an example of the potential general applicability of our method.

The agent in `swimmer-v2` is a snake-like robot that can be controlled by actuating its two joints. This is in contrast with environments from `meta-world` and `point-maze`, in which the agent's position can be directly controlled. Moreover, instead of extracting expert trajectories through a scripted policy, we use a policy trained from scratch with DDPG (Lillicrap et al., 2019). These trajectories are then used to train behavioral and temporal priors, as described in Section 5.3. We then train our method, as well as all baselines on two downstream task. In `swimmer-sparse`, dense rewards are replaced with a binary reward signal that is only positive when the agent is farther than a sparsity factor $\lambda_s = 1.5$ from its initial position. `swimmer-sparse-noisy` also adopts this sparse reward scheme and, additionally, corrupts observation by constant additive noise on the state $s_t = s_t + \epsilon$ with $\epsilon = 0.5$. This can be interpreted as bad calibration of sensors or faulty readings from a real-world robot. Results for this experiments are reported in Figure 11.

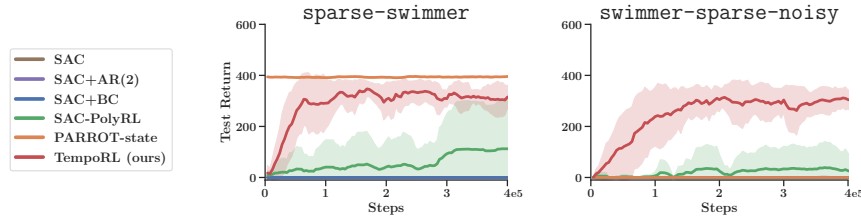

**Figure 11:** Performance on tasks derived from `swimmer-v2`. Temporal priors are capable of guiding exploration even when states are corrupted by noise.

We observe that, as expected, in `swimmer-sparse` PARROT is very effective, since training and downstream environments are identical. TempoRL is also able to learn a successful policy, albeit at a slower rate, while the remaining baselines fail to observe any meaningful reward. However, when noise is introduced in the observation space, TempoRL still retains good performance, while PARROT fails to transfer to this unseen task.

## F.4 MODEL ABLATION

We finally set out to provide empirical backing for an important design choice. Our policy mixing approach grants freedom in choosing generative models capable of describing complex distributions, as computing distance metrics to the prior is not required. We compare the performance of Real NVP Flows with a Conditional VAE (Sohn et al., 2015), a non-parametric conditional Least Squares Density Estimator (Sugiyama et al., 2010), and a MLP modeling a deterministic prior. As shown in Figure 12, we observe that Flow modeling is consistently more powerful.

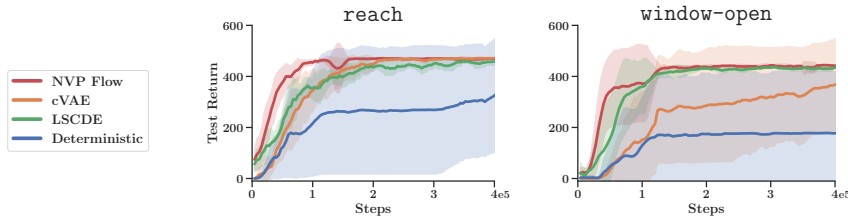

**Figure 12:** Performance on downstream tasks when modeling a one-step state-independent temporal prior through various generative models.

## F.5 LEARNING ALGORITHM ABLATION

As described in Appendix D.3, HER (Andrychowicz et al., 2017) and n-step returns are the two main algorithmic modifications that we introduce on top of the SAC framework (Haarnoja et al., 2018). As our method significantly benefits from these two components, we also apply them to all baselines to ensure a fair comparison across our experiments.

In this section, we provide a concise discussion on the importance of these two modifications in the form of an ablation study. We compare the performance of TempoRL on `reach` and `window-open` when i) disabling HER and ii) changing the number of steps for reward computation from 10 to 5, 2 and 1. In Figure 13, we can observe that n-step rewards are a crucial component to performance in long-horizon tasks, as they can effectively accelerate the propagation of value to states that are distant from the goal. On the other hand, we find HER's contribution to be marginal, and counterproductive in certain settings (e.g. in `reach`) .

## F.6 STUDY OF SAC+AR(N)

SAC+AR(n) can be a strong baseline, even when compared to most elaborate methods. However, the value of $n$ is crucial to final performance and it requires extensive tuning or domain knowledge. In this section, we present a brief study of its performance on the `meta-world` tasks. We observe that, within the experimental settings outlined in Section 5.3, larger values of $n$ can help in some tasks (e.g. `door-close`). However, they are not sufficient to solve the harder tasks and can be detrimental when precise control is required, as in `reach`.

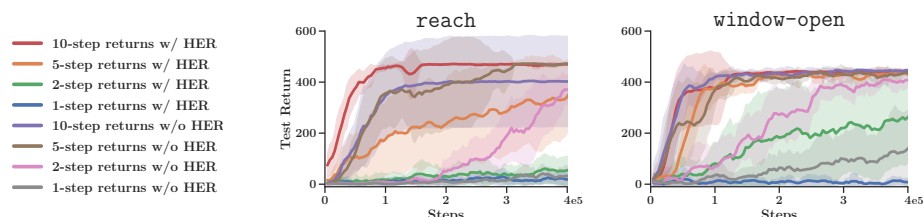

**Figure 13:** Performance on downstream tasks when disabling or enabling HER, with different horizons for n-step returns.

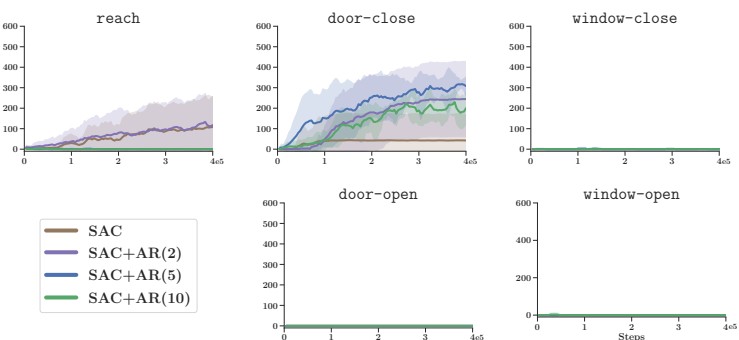

**Figure 14:** Performance on SAC+AR(n) for different values of $n$.

## F.7 CORRELATION AND STATE COVERAGE

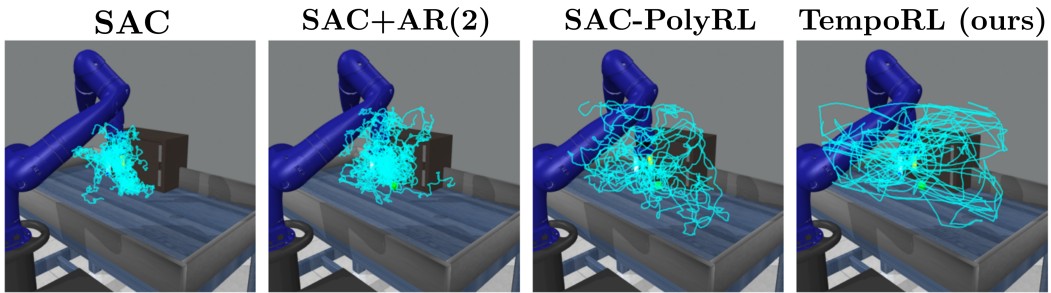

**Figure 15:** A qualitative comparison of sampled exploration trajectories in a robot manipulation environment (`reach`). Our method achieves directed behavior while covering most of the state space, outperforming SAC-PolyRL. On the other hand, uniform sampling (SAC) and action-repeat (SAC+AR(2)) fail to cover the full state space.

## F.8 VISUAL RL

We report the performance on an additional unseen task (`window-open`) in the same settings as in Section 5.3. Similarly, our method outperforms both PARROT and SAC (see Figure 16), which are not able to learn the task.

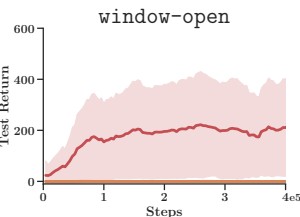

**Figure 16:** Comparison of methods on `window-open` with image-based states: TempoRL (red), SAC (brown) and PARROT (orange).

## G ADDITIONAL RELATED WORKS

**Meta-RL and Task Inference** The idea of quickly adapting to unseen tasks is a fundamental concept in meta-reinforcement learning (Schmidhuber, 1987; Wang et al., 2017). While TempoRL also

aims at tackling unseen tasks, it crucially relies on training a policy from scratch for downstream learning and is not designed for zero-shot adaptation. A line of research in meta-RL, referred to as *context-based* (Wang et al., 2017; Mishra et al., 2018; Rakelly et al., 2019), performs task inference in order to identify the current task and quickly extrapolate how to maximize returns. A task representation can in practice be extracted from recent experiences.

Interestingly, temporal priors can in principle handle the same type of input, that is sequences of recent action-state pairs. However, instead of producing an explicit representation of the current task, temporal priors directly model an action distribution. Nonetheless, while in the context of this paper we only train temporal priors on single tasks, they could in principle be trained on multiple tasks, and learn to perform implicit task inference on the current trajectory in order to produce promising actions.

