# OpenReview forum: "TempoRL: Temporal Priors for Exploration in Off-Policy Reinforcement Learning"
_ICLR.cc/2022/Conference — ICLR 2022 Submitted_

### Official Review · Reviewer_66v7 · 2021-10-26

**Correctness:** 3
**Technical Novelty And Significance:** 2
**Empirical Novelty And Significance:** 3
**Recommendation:** 3
**Confidence:** 4

**Details Of Ethics Concerns:**

There are no ethical concerns.

**Main Review:**

The proposed solution is benchmarked against a rather large set of related approaches and it is shown to perform very well, for tasks that differ significantly from the original tasks in particular. It also allows for the exploration of larger actions spaces without sacrificing the ability to achieve high rewards.

It is shown that conditioning the prior on the current state complicates transfer to tasks that differ too much from those of the source domains. What is clearly shown in the paper is that you often benefit from instead ignoring the state and only relying on the recent actions. Of particular interest is the fact that a prior only conditioned on the most recent action, rather than a longer sequence of actions, is as competitive as it is.

Unlike behavioural priors, the proposed temporal do not assume that the state space has yet been explored, which in turn would assume that the novel tasks need to be similar to those the prior was trained on. Another benefit of the proposed priors is that they can be trained off-line in many different ways, such as by exploiting policies from earlier tasks or from demonstrations.

However, the most important concern when it comes to the proposed approach is the inclusion of the mixing weights in (4). It is hard to see where this actually comes from and it would be good for the authors to sort this out. The addition would suggest the entropy component is suppressed for samples drawn from the policy and is instead dominated by those drawn from the prior, which should further encourage the policy to explore a wider space of actions, which might explain the good performance. Another concern is that since the samples are drawn from the mixture you no longer have an approximation of an entropy. It seems more like a cross-entropy between the mixture distribution and the policy distribution.

The addition of the monotonically increasing function with parameters that have to be experimentally set also seems like an arbitrary solution that needs a better explanation. The reasoning behind the function is quite unclear in the paper. If the mixing weights in the objective are seen as a kind of importance sampling, then the expectation would not be over a distribution that sums up to 1, which might motivate the need for the monotonic function. It is true though that without the inclusion of the mixing weights in the objective, the proposed solution becomes very similar to Soft Actor-Critic, with the main difference being the sampling.

The notation a_t ~ pi(s_t) in (2) and (3) is a bit confusing since it gives the impression that the policy could be deterministic. It would be better to include a do, as in pi(.|s_t). Further, since policy parameters are used in (5) and (6), it would be good to include also the parameters for the Q-function. By the way, the parameters for the Q-function are called both phi and psi in the paper. The paper does not mention how alpha is controlled over time, which can be understood from earlier literature but would be good to include also here.

Finally, from the experiments, it is hard to deduce whether observed improvements are due to the mixed sampling or from the changed policy objective that includes the mixing weights. It would be good to test with and without the weights, assuming there is no good motivation why the weights need to be in the objective.


**Summary Of The Paper:**

Temporal priors for exploration in off-policy reinforcement learning are proposed, priors for action sampling that are state-independent and depend only on the most recent past actions. In essence, an earlier trained temporal prior come in as a replacement, if a policy shows a great deal of uncertainty at a particular, possible yet unexplored, state. Exploration resorts to more generic task-agnostic trajectories instead of trusting the current policy. It is shown that sampling from such a mixture of policy and temporal priors can accelerate transfers to novel tasks that differ significantly from those originally trained for.

**Summary Of The Review:**

Given the concerns raised in the core part of the paper that relates to the policy objective, the paper cannot be directly accepted, but not completely rejected either. There might be a good motivation behind the objective that the authors may be able to clarify, something that would lead to a minor modification of the paper.

---

> ### Author Response · Authors · 2021-11-20
> **Response to 66v7**
>
> We thank the reviewer for the detailed assessment. Our answers are presented below.
>
>
> **Eq. 4 and mixing function**
>
> We have adapted the notation in the paper for more clarity. We will now motivate our objective more precisely.
>
> Intuitively, SAC’s regularization encourages the policy’s entropy to grow, which results in sampling more diverse actions and improves exploration.
> As we show in Section 5.2, sampling actions from a temporal prior can produce trajectories that cover the state space more efficiently with respect to a max-entropy (uniform) distribution.
> Thus, instead of sampling from the policy alone and increasing its entropy, it would be more desirable to sample from a mixture between policy and prior, and increase the likelihood of sampling from the latter.
>
> As a consequence, we *replace* the entropy term $\mathcal{H}(\pi(\cdot|s))$ in SAC with a new term $\lambda=\Lambda(\mathcal{H}(\pi(\cdot|s)))$. $\lambda$ represents the mixing weight, or the likelihood of sampling from the prior. We note that $\Lambda$ does not represent a multiplicative factor for the entropy, but a mixing function. The output of the mixing function $\lambda = \Lambda(\mathcal{H}(\pi(\cdot|s)))$ replaces the entropy term entirely in eq. (4).
>
> A separate argument needs to be made for how the mixing weight is then computed. In principle, this could happen in a number of ways.  In practice, we define the mixing function $\Lambda(\cdot)$ as a fixed monotonic function of an entropy estimate.
> This approach was chosen to reduce the complexity of our method. While the mixing function could in principle be parameterized and learned, our solution is considerably simpler and leads to good empirical performance.
>
> The particular design of $\Lambda$ as a scaled sigmoid $\Lambda(x) = \sigma (\beta_s x + \beta_t )$ is naturally motivated by two observations:
> (1) intuitively, when the policy cannot maximize the reward signal, its entropy will grow and it will be beneficial to sample more often from the temporal prior, therefore the output of the mixing function should be monotonic with respect to its input
> (2) as the output will represent a mixing weight, it should belong to the interval [0, 1].
> A widely used function that respects these two observations is the sigmoid. We add scaling parameters $\beta_s, \beta_t$ to increase its flexibility.
>
> **Ablation of learning objective**
>
> Modifying the objective to include mixing weights explicitly can potentially describe methods in which the mixing weight is not directly computed from the entropy. Since, in our case, the mixing weight is a monotonic function of the entropy estimate, our proposed regularization term and the one proposed in SAC have the same global minimum. Optimizing the original objective therefore remains a viable strategy, as suggested.
> We offer an extensive comparison between the performance of SAC-TempoRL with a variant which optimizes the unmodified SAC objective in Appendix B. We empirically observe that modifying the objective leads to improvements in performance.
>
> **Entropy regularization coefficient ($\alpha$)**
>
> Our method can be augmented with automatic entropy tuning in a straightforward way, as described in [1]. For simplicity, within the scope of this paper, $\alpha$ is kept fixed for all methods, as initially proposed in [2]. We modified Appendix D.3 to put more emphasis on this remark.
>
> **Notation**
>
> We changed the notation in Section 4.2 according to the reviewer’s suggestions. Parameter vectors were included in the Q-function and inconsistencies were fixed. We thank the reviewer for the pointer.
>
> **References:**
>
> [1] Haarnoja, Tuomas et al. Soft Actor-critic Algorithms and Applications. 2018
>
> [2] Haarnoja, Tuomas et al. Soft Actor-critic: Off-policy Maximum Entropy Deep Reinforcement Learning with a Stochastic Actor. ICML 2018

---

> > ### Comment · Reviewer_66v7 · 2021-11-22
> > **Response to rebuttal**
> >
> > We thank the authors for addressing concerns raised in the revised manuscript with its appendix and clarifications made in the rebuttal.

---

> > > ### Author Response · Authors · 2021-11-23
> > > **Re: Response to rebuttal**
> > >
> > > We thank the reviewer for the feedback and for considering our clarifications.

---

> > > ### Author Response · Authors · 2021-11-26
> > > **Additional Feedback**
> > >
> > > We have noticed that the reviewer has updated the score since the discussion board has been re-opened. We politely inquire if there are any unresolved questions or remaining concerns that we could provide feedback on. We thank the reviewer in advance for the discussion.

---

### Official Review · Reviewer_wMYF · 2021-11-01

**Correctness:** 2
**Technical Novelty And Significance:** 2
**Empirical Novelty And Significance:** 2
**Recommendation:** 3
**Confidence:** 4

**Main Review:**

Strengths:
1. The paper tackles a difficult but prevalent problem in RL: solving control environments with long horizons and sparse rewards.
2. The proposed learned priors showed good transfer properties on selected environments, some even with a wide gap.

Weaknesses:
1. The baselines being compared to are rather weak both in terms of their exploration capabilities, and that they were not designed with transfer tasks in mind. There seem to be many other baselines that could be considered in this space, such as intrinsic motivation based agents (e.g. Agent57), randomized value function / policy based exploration, or model-based exploration agents (e.g. Plan2Explore) etc.
2. There is danger of overfit to selected control environments when the paper's statements for contribution seem much broader. After reading the paper, I'm not sure why such temporal priors would necessarily generalize onto other long horizon sparse reward environments. Is there a piece of theory, an explanation or an illustration that could help convince us of the general usefulness of these temporal priors?

**Summary Of The Paper:**

The paper proposed learned temporal action priors conditioned on past trajectories of the agent, and demonstrated using it in a soft actor critic agent, applied to long-horizon control problems.

**Summary Of The Review:**

The paper presents a set of temporal priors for exploration, empirically showing good results for certain transfer tasks. I'm concerned with the generality of the approach beyond the selected environments and the lack of competitive baselines that the paper considered.

---

> ### Author Response · Authors · 2021-11-20
> **Response to wMYF**
>
> We thank the reviewer for the feedback. We would like to (a) motivate our choice of baselines and (b) discuss the general applicability of our method and limitations.
>
> **(a)** Our method enables correlated exploration by sampling from a learned temporal prior. For this reason, we believe that the most related methods are those targeting correlated exploration (SAC+AR, SAC-PolyRL), or those that attempt to overcome exploration by transferring knowledge from expert trajectories (PARROT, SAC+BC). This choice of baselines is consistent with those proposed by previous works [1, 2] in this field.
>
> While Agent57 [3], RND [4] and Plan-to-explore [5] are powerful approaches to exploration, we argue that they are not strictly relevant to our setting, because they (1) do not aim at transferring knowledge and (2) do not directly address temporally consistent behavior. Moreover, these three methods are proposed for a visual RL setting.
>
> For this reason, we have given priority to testing the more-closely related baseline proposed by reviewer [eq2m], which we extensively experimented with. In short, we have found that a hierarchical approach helps in this setting, but learning skills from offline trajectories represents a fundamental obstacle.  For more detailed results, please see the answer to the respective review. We would like to ask if the reviewer agrees with this choice.
>
> **(b)** Throughout the paper, we support the general usefulness of temporal priors, within the setting we propose, by explaining how current methods fail to address challenges that come with sparse rewards, long horizon tasks and availability of task-agnostic demonstrations. Moreover, our arguments are backed up by an empirical evaluation showing how temporal priors can be a viable solution. In the context of the rebuttal, we worked on providing additional empirical evidence by evaluating our method in more tasks. These results were added in Appendix F.
>
> We acknowledge that our method does not represent a general solution to exploration in reinforcement learning, and that methods leveraging additional assumptions can outperform it in certain settings. We emphasize, however, that in the setting proposed, namely the transfer to related but unseen tasks, TempoRL poses a viable solution.
>
> Another important consideration is that, while state-independent temporal priors are able to generalize across larger domain gaps with respect to baselines, they can only be as general as the data used for training. In other words, temporal priors are data-driven, and as such will only reconstruct behavior that appears in the offline dataset of trajectories they are trained on. For instance, when training on `reach` with random goals, temporal priors will not encode any grasping strategy. While such behavior can still be recovered by the policy $\pi$, there is no incentive to do so. As a result, successfully transferring to tasks that require additional actions such as grasping (e.g. `pick-place`) remains an open challenge, as we show in Appendix F.1.
> This discussion of limitations was extended and added as Appendix E. We remain open for suggestions regarding content that the reviewer feels could provide additional backing to our method.
>
>
> **References:**
>
> [1] Singh, Avi et al. PARROT: Data-driven Behavioral Priors for Reinforcement Learning. ICLR 2021
>
> [2] Amin, Susan et al. Locally Persistent Exploration in Continuous control Tasks with Sparse Rewards. ICML 2021
>
> [3] Badia, Adrià Puigdomènech et al. Agent57: Outperforming the Atari Human Benchmark. ICML 2020
>
> [4] Burda, Yuri et al. Exploration by Random Network Distillation. ICLR 2019
>
> [5] Sekar, Ramanan et al. Planning to Explore via Self-Supervised World Models, ICML 2020

---

> ### Comment · Reviewer_wMYF · 2021-11-23
> **reply to author responses**
>
> Thank you for uploading the additional results and all the other modifications. For an empirical method like the one proposed in the paper, I fail to see strong evidence of generality in the learned priors across a wide range of environments despite of the good improvement on the paper during rebuttal. The additional tasks from MT10 show no or noisy improvements from the baselines. There is only one environment selected from Mujoco and the complex maze is not qualitatively different from the U maze.

---

> > ### Author Response · Authors · 2021-11-26
> > **Re: reply to author responses**
> >
> > We respectfully suggest that our empirical results are consistent with our claims on generality, in particular when considering the additional experiments provided during the rebuttal phase.
> >
> > MT10 was designed to include a diverse set of tasks [6]. While the baselines only succeed in the 3 simplest tasks from `meta-world`, TempoRL can solve 8/10. The remaining 2 tasks (`pick-place` and `push`) involve grasping, which does not appear in the dataset of trajectories and cannot be encoded in the learned prior, as discussed in Appendix F.1 and Appendix E. We note that the 2 tasks in which TempoRL's performance has larger variance (`drawer-open` and `button-press`) require very precise control of the end effector, and remain out of reach for all the considered baselines.
> >
> > The additional maze structure we experimented on was designed to address concerns on generality. For this reason, the maze layout includes dead ends and crossroads, and the optimal solution features a greater number of turns with respect to `u-maze` (5 vs 2). For the purpose of further experimentation, we ask the reviewer what maze designs would represent a stronger argument with respect to generality.
> >
> > **References:**
> >
> > [6] Yu, Tianhe et al. Meta-world: A Benchmark and Evaluation for Multi-task and Meta Reinforcement Learning. CORL 2020.

---

### Official Review · Reviewer_txKn · 2021-11-01

**Correctness:** 4
**Technical Novelty And Significance:** 2
**Empirical Novelty And Significance:** 3
**Recommendation:** 6
**Confidence:** 3

**Main Review:**

Overall, the empirical results are impressive, but I feel that the potential weakness of the method is neither tested nor discussed.

# Strength
- S1. The empirical results of the proposed method are impressive.
- S2. It is nice that the proposed method is agnostic of the state space of the downstream task.

# Weakness

The detailed comment is given in the next section.

- W1. The paper seems to claim that previous methods suffer from the domain gap while the proposed method does not. However, I can think of some situations that this method could be problematic.
- W2. This is somehow related to W1. I feel that the paper tries to overly sell the proposed method.
- W3. The objective function for the proposed SAC-like algorithm contains a mistake.

# Detailed Comments

As for W1, it seems to me that the proposed method can suffer from the domain gap in some cases. For example, let us consider a maze
(not a simple one like U-maze) with visual state information, i.e., the agent can observe the maze from bird's-eye view. Then, it is likely that without conditioning the prior by a state, the agent might simply hit the wall and continue to do so because of the state-agnostic prior. To claim that the proposed method does not suffer from the domain gap, would you add experiments in an environment like a general maze?

In addition, from Figure 5, I have an impression that the proposed method learns to simply repeats the same action, occasionally changing it. It seems that all downstream tasks are solvable with this simple strategy since the agent can occasionally hit the door or window knob, resulting in opening/closing it. Would you provide results of SAC+AR(n) with a higher n, like 5, 10, or so.

As for W2, for example, the paper says that "... we propose a principled manner of integrating priors..." What do you mean by "principled"? The proposed algorithm seems to use an ad-hoc way to adjust the ratio of a mixture policy. (Furthermore, the derivation of the proposed objective seems to contain a mistake.)

Also, the paper says "We show how state-independent temporal priors can be learned from few expert trajectories on simple tasks and used to improve exploration efficiency in new tasks, despite the presence of massive domain gaps". How do you measure the size of domain gaps? To me, a reaching task and a window opening task are not so different. In case of a reaching task, a robot needs to move its arm to the direction of a target, whereas in case of a window opening task, a robot similarly needs to move its arm to the direction of window opening while pushing the knob of the window. If a downstream task is grasping and stacking objects, I would say it is completely different, though.

There are many other claims which I feel is not well supported, and this is confusing to me.

As for W3, the maximization of the objective (Eq 4) cannot be achieved by the minimization of Eq 5. To see this, let us look at Eq 11 and Eq 13. If you take the expectation of Eq 13 (ignoring $\min$) wrt the next state, reward, and next action, what you will get is $\mathbb{E}[r + \gamma (1-d) Q(s', a') + \alpha \Lambda (-\log \pi (a'|s'))]$, which is not equal to Eq 11 unless $\Lambda$ is a linear function. If $\Lambda$ is a linear function, the proposed objective is the same as SAC's objective. In other words, Eq 13 is NOT an unbiased estimate of Eq 11. The same argument can be made about Eq 4. I think the biasedness is OK as long as the paper clarifies this, and the algorithm still works. But it needs to be clarified.

# Suggestions

In sum, I suggest the authors to improve the draft by removing some overly claimed sentences and adverbs, add some more experiments with downstream tasks which require completely different skills like grasping, and consider pointing out the biasedness of derived estimators.

# Post-discussion

After the author rebuttal, I am mostly satisfied with it and raised the score to accept. While I am still OK to accept the paper if the authors discuss the limitation of the method and correct some overselling wordings in the main text, I was convinced that my score should be based on what is in the current pdf not the paper's potential. Therefore, I lowered the score to weak accept. I strongly encourage the authors to resubmit the paper with discussion on the limitation of the method and correction of some overselling wordings in the main text.

**Summary Of The Paper:**

This paper considers the exploration problem in RL and proposes a new type of the behavioral prior, one only conditioned by recent actions. The paper empirically demonstrates the effectiveness of this approach in sparse-reward environments. It also shows that this type of the behavioral prior can be learned in a simple non-visual task and transferred to a visual task.

**Summary Of The Review:**

The paper proposes an interesting idea, and current experimental results are impressive. However, I think the proposed method is not well-tested or discussed deeply. Accordingly, I am more inclined to the rejection unless my concerns are resolved.

---

> ### Author Response · Authors · 2021-11-20
> **Response to txKn**
>
> We are thankful for the detailed review and set out to address each concern that the reviewer raised.
>
>
> **General mazes**
>
> There are settings in which our method can also suffer from domain gaps (see Appendix E). In the paper we empirically argue that TempoRL is noticeably more robust to domain gaps, and, when using state-independent temporal priors, indifferent to changes in the state space.
> We understand the concern with more complex maze structures. When only having access to expert trajectories on a simple maze (`room-maze`), hitting walls in a complex maze is an issue that TempoRL and all baselines would face. In order to overcome this, some assumptions about the environment would need to be taken (e.g. as is done in [1]), which would result in restricted applicability.
> While it does not directly address this issue, we found that our method does fairly well on complex maze structures that are beyond the reach of baselines. The results were added as Appendix F.2.
>
>
> **Action repeat**
>
> SAC+AR(n) can be a strong baseline, even when compared to more elaborate methods. However, the value of $n$ is crucial to final performance and it requires extensive tuning or domain knowledge. We provide a comparison for different action-repeat values in Appendix F.6. We found that increasing the value of $n$ improves performance in some task, but can be counterproductive in tasks that require precise control (e.g. `reach`).
>
>
> **Tone of the paper**
>
> We have clarified our claims in the sections mentioned and will provide additional justifications here. We would be happy to address any other sentence which the reviewer considers to be an overstatement.
>
> - We had defined our approach as *principled* because we integrate our sampling method in the learning algorithm. In order to bring more clarity, we have  removed the statement accordingly.
>
> - In general, we use an informal definition of *domain gaps*. We borrow this term from visual adaptation and do not attempt to formally define it, but rather use it as a motivation for our work. The domain gap between two tasks could be quantified by how well a policy trained in environment A would perform in environment B after fine tuning. If the MDP governing environment A is similar to that governing B, the learning agent should perform well in environment B, which can be interpreted as a sign of a small domain gap. If the state space or action space or dynamics vary significantly, the domain gap can be seen larger. We note that this measure might attribute large domain gaps to pairs of tasks that are quite similar from a human perspective, as a consequence of the notorious brittleness and difficult generalization of RL compared to, for instance, supervised learning. This answer has been summarized and added to Footnote 1.
>
> - Finally, we have added a discussion of limitations in Appendix E. In particular, we describe how the effectiveness of temporal prior depends on the nature of the offline dataset of task-agnostic trajectories used for training. Moreover, we address how, despite generalizing well to some tasks, our method can also suffer from large domain gaps. For example, when trained on simple reaching tasks, temporal priors would not encourage the agent to attempt precise grasping strategies, and can thus be inefficient at guiding exploration in environments that require this.  For an extended discussion on these topics and additional challenges, we refer the reviewer to Appendix E.
>
> **Biased estimate of eq. 11**
>
> We corrected the text to explicitly state that our estimate is biased. We thank the reviewer for pointing out this important clarification.
>
>
> **References:**
>
> [1] Pitis, Silvu et al. Maximum Entropy Gain Exploration for Long Horizon Multi-goal Reinforcement Learning. ICML 2020

---

> > ### Comment · Reviewer_txKn · 2021-11-22
> > **Re: Response to txKn**
> >
> > Thanks a lot for the revision and response. Additional experiments and discussion about the limitation of the proposed method resolved my concerns. Accordingly, I raised my score.

---

> > > ### Author Response · Authors · 2021-11-23
> > > **Re: Re: Response to txKn**
> > >
> > > We thank the reviewer for considering our revision and for increasing the score.

---

### Official Review · Reviewer_eq2m · 2021-11-02

**Correctness:** 3
**Technical Novelty And Significance:** 3
**Empirical Novelty And Significance:** 2
**Recommendation:** 3
**Confidence:** 3

**Main Review:**

__Strengths__:

1. The paper is well-written and easy to follow.

2. The authors provide an appropriate and concise overview of the most related prior work and its limitations.

3. The proposed idea of capturing correlation between different actions is simple, intuitively useful, and general since it only requires the action space to match between the 'training' and 'downstream' environments.

4. The authors provide experimental results analyzing different aspects of temporal priors, such as the effects of the conditioning variables and the type of employed generative model.

__Main Concerns/Doubts__:

My major concerns with this work are the objective formulation and the experimental results. In order of relevance:

1. Both hindsight experience replay (HER) and n-step returns are prior techniques that are orthogonal to the author's contribution. Both applicability and effectiveness of these techniques are very much environment dependant (e.g. HER requires access to the true reward function). Hence, it would be important if the authors could clarify whether these are also adopted by the other baselines. Moreover, it would be very relevant to include an ablation study to test the contribution of these ancillary methods and the effectiveness/sample-efficiency of TempoRL without them.

2. The authors benchmark their algorithm on a very small, handpicked selection of adapted environments. For a more comprehensive and fair evaluation for robotics manipulation, I believe it would have been more appropriate to consider all of the environments in one of the pre-specified evaluation modes in meta-world (e.g., MT10). Thus, it would be important to either include additional experiments or, at least, provide a justification for the specific environment choices.

3. It is unclear what is the effect of the proposed optimization objectives (Equations 4-7) on the sampling distribution policy. The authors propose to use a sampling policy derived from a mixture of $\pi$ (the parametric downstream policy) and $\bar{\pi}$ (the temporal prior). Hence, to directly optimize w.r.t. the sampling distribution (as in SAC), both the policy's optimization objective and the targets for the Q-value would need to sample actions from the same mixture distribution. Yet, both of these objectives inconsistently consider sampling actions solely from $\pi$.  If possible, It would be important to justify this specific choice over its natural alternative through formal analysis of these objectives.

4. The baseline algorithms utilized have been proposed for different problem settings:
 - PolyRL (Amin et al. 2020) does not make use of prior trajectory data.
 - PARROT (Singh et al. 2020) assumes the prior trajectory data comes from a diverse set of multiple training environments from the same distribution as the downstream environment.
  -> It would be appropriate to compare TempoRL to additional methods/algorithms that have been designed for reinforcement learning transfer given experience/demonstrations from a single environment, since this problem setting more closely matches the examined one. For instance, the authors could consider algorithms that learn diverse low-level skills in a training environment. E.g. the work from Florensa et al. 2017 and/or Haarnoja et al. 2018 (both of which share their code) could be adapted to use offline data by treating each trajectory as representing a different skill.
 -> This work asserts itself as removing the fundamental limitations of behavioral priors (Introduction, 3rd paragraph), without analyzing its own introduced limitations. Hence, for a more comprehensive comparison with behavioral prior algorithms, such as PARROT, it would be important to perform evaluation also on the behavioral priors problem setting. For instance, this could be done by adding an experiment treating all environments but one as training environments. This would assess the scalability of the proposed algorithm by evaluating how much useful information can be extrapolated from the correlation of consecutive actions in trajectories from multiple diverse training environments.

5. The authors introduce a conspicuous amount of additional hyperparameters without providing details on how they were selected or an ablation study that provides information about the algorithm's robustness to their choices (e.g., $\alpha$, $\beta_t$, $\beta_s$, hindsight replay ratio, n-step, ...).

__Minor concerns__:

1. Given the high standard deviations for some of the performance curves, I believe the authors should increase the number of utilized random seeds from the current 3/5 per experimental setting (Peter et al. 2018).

2. It is unclear to me how the policy's entropy quantifies the agent's confidence to reach the goal, as stated in Section 4.2 (if multiple actions lead to the same outcome, wouldn't the entropy still be high regardless of the agent's confidence?).

3. Figure 5 does not show the exploration results for Parrot.

4. which is capable -> that is capable (Introduction, 3rd paragraph).

__Further suggestions__:

1. It would be interesting to compare the proposed SAC integration with utilizing the temporal prior as a reparameterized action space, as done in many hierarchical RL methods and PARROT.

2. Since incorporating past states and actions provides information about the underlying task, a further connection could be drawn between the concept of temporal priors and implicitly performing task inference (e.g. as in Rakelly et al. 2019).


__Post-rebuttal__

While I appreciate the additional experiments performed by the authors, I believe they accentuate very severe issues w.r.t. the paper's empirical results. For instance, there is a large gap between the results in the original hand-picked meta-world tasks and the other tasks in MT-10. Moreover, there is no formal analysis to provide strong intuition about the algorithm's exploratory properties. This would be particularly important given some of the authors' algorithm design choices, which appear to be inconsistent with prior literature (not optimizing policy and Q-function w.r.t. exploratory policy).

While I like the simple idea introduced in the paper, after reading the rebuttal and the other reviews, I do not think that TempoRL is currently shown to make a significant contribution. Thus, I think this paper is not yet ready for publication.


__References__

Yu, Tianhe, et al. "Meta-world: A benchmark and evaluation for multi-task and meta reinforcement learning." Conference on Robot Learning. PMLR, 2020.

Amin, Susan, et al. "Locally Persistent Exploration in Continuous Control Tasks with Sparse Rewards." arXiv preprint arXiv:2012.13658 (2020).

Singh, Avi, et al. "Parrot: Data-driven behavioral priors for reinforcement learning." arXiv preprint arXiv:2011.10024 (2020).

Florensa, Carlos, Yan Duan, and Pieter Abbeel. "Stochastic neural networks for hierarchical reinforcement learning." arXiv preprint arXiv:1704.03012 (2017).

Haarnoja, Tuomas, et al. "Latent space policies for hierarchical reinforcement learning." International Conference on Machine Learning. PMLR, 2018.

Henderson, Peter, et al. "Deep reinforcement learning that matters." Proceedings of the AAAI conference on artificial intelligence. Vol. 32. No. 1. 2018.

Rakelly, Kate, et al. "Efficient off-policy meta-reinforcement learning via probabilistic context variables." International conference on machine learning. PMLR, 2019.



**Summary Of The Paper:**

This paper proposes the concept of temporal priors, extending previously proposed behavioral priors. Behavioral priors try to recover some prior (state conditioned) policy utilizing a set of trajectories obtained from multiple 'training' tasks, to be used to guide exploration in 'downstream' tasks. Instead, temporal priors take a more general non-Markovian approach by learning a prior policy that is potentially conditioned on the whole history of previous states and actions. This work proposes a new algorithm, TempoRL, that employs a generative flow policy to parameterize the temporal prior and a modified SAC agent that is incentivized to perform exploration with the temporal prior. In practice, the authors propose to limit the conditioning information of the temporal prior to, exclusively, the previous action in the trajectory, only learning the correlation between consecutive actions. This choice makes the resulting temporal prior less dependent on the diversity and the optimality of the trajectory data from the training tasks than traditional behavioral priors. Empirically, the proposed algorithm obtains promising results in learning temporal priors from a single training task in a few robotics environments from meta-world (Yu et al. 2020) and simple navigation environments with sparse rewards.

**Summary Of The Review:**

While this paper introduces some simple and interesting ideas, without a more comprehensive experimental evaluation and a clearer explanation of the properties of the algorithm, it is not yet ready for publication.

---

> ### Author Response · Authors · 2021-11-20
> **Response to eq2m (1/2)**
>
> We thank the reviewer for the many suggestions and detailed assessment. A detailed response follows.
>
>
> **HER and n-step returns**
>
> All baselines we adopted are also using HER and n-step returns to ensure a fair comparison. We modified Appendix D.3 to explicitly state this. Furthermore, we ran the requested ablation study  and provide the results in Appendix F.5. In short, we found that n-step returns are key to good performance in long-horizon settings, as they can effectively propagate rewards across many time steps. On the other hand, HER only contributes marginally to sample efficiency and final performance.
>
>
> **Additional meta-world tasks**
>
> Our selection of meta-world tasks is meant to keep the evaluation concise. We select the (arguably) easiest task for training (`reach`) and all tasks related to two representative objects (door and window) as downstream tasks.
> As requested, we ran additional experiments on all remaining MT10 tasks. We found that the additional results (reported in Appendix F.2) are consistent with our claims.
>
>
> **Additional baselines**
>
> We thank the reviewer for the suggestions about additional baselines. The baselines we compare to are designed to tackle the same problem, that is correlated and temporally extended exploration. However, to the best of our knowledge, no viable method has been proposed to leverage an offline dataset of task-agnostic demonstration. Among the baselines we have chosen, even those that can in principle utilize expert data were found to struggle when dealing with task-agnostic trajectories or large domain gaps (Section 5.3).
>
> Among the two baselines mentioned, we have worked on adapting [1] to our setting, as it directly deals with transferring knowledge across a domain gap. As the reviewer correctly anticipates, this baseline needs to be adapted, as it originally requires access to the training environment, while our setting assumes access to offline trajectories.
> For this reason, while the low-level policy $\pi(a|s, z)$ does not need modification, the skill learning algorithm needs to be adjusted. In order to adapt to the off-policy scenario, TRPO can be replaced by SAC. Still, the proposed method requires labeling each trajectory with a categorical skill label. Due to the number of trajectories for training, labeling each one differently would produce an impractical amount of skills. We therefore experimented with uniformly sampling labels or selecting labels that maximize mutual information between states and labels (see [1], Section 5.3) through iterative optimization. Additionally, we attempted to train skills by using behavioral cloning instead of offline RL. All of these attempts resulted in low-quality skills which prevent the high level policy from reaching any meaningful return when trained on downstream tasks. We note that the high-level policy did not need any meaningful modification, but in practice we replaced TRPO with a discrete version of SAC in order to improve sample efficiency.
>
> In conclusion, we believe that the baseline presented in [1] fails to perform well within our setting due to its inability to learn meaningful skills from offline data without major modifications or hand-crafted skills.
> In fact when providing the high-level policy with fixed skills (each one moving the agent in each of the 6 cardinal directions in a 3D space), the method can perform well (see Table below). This however involves manually designing skills.
> We believe that the second proposed baseline [2] incurs similar issues.
>
> ```
> |       |  reach-v2  | window-open-v2 | window-close-v2 | door-open-v2 | door-close-v2 |
> |  (A)  |    0+-0    |      0+-0      |      0+-0       |     0+-0     |      0+-0     |
> |  (B)  |    8+-12   |      0+-0      |      0+-0       |     0+-0     |      3+-6     |
> |  (C)  |  305+-45   |    373+-27     |    416+-3       |     0+-0     |    419+-10    |
> |TempoRL|  471+-1    |    443+-5      |    443+-11      |   291+-184   |    445+-1     |
> ```
> Table 1: Test returns at 500k steps (5 seeds). (A) is an adaptation of [1] that learns skills through offline RL; (B) is an adaptation of [1] that learns skills through behavior cloning; (C) is an adaptation of [1] that uses hand-crafted skills.

---

> > ### Author Response · Authors · 2021-11-20
> > **Response to eq2m (2/2)**
> >
> >
> > **Effect of optimization objectives (eq. 4-7) on the sampling distribution policy**
> >
> > In our framework, we do not optimize the sampling policy used for data collection (which is a mixture between the prior $\bar \pi$ and policy $\pi$). While following a temporal prior is beneficial for exploration during training, at test time actions are directly sampled from the policy alone (mixing weights are set to 0). For this reason, learning objectives are directly optimized with respect to the policy $\pi$. In practice, the prior $\bar \pi$ only comes into play for data collection, while optimization only trains the (current) policy $\pi$, as is done in SAC. In case this intuitive explanation does not address the reviewer’s concerns, we would  kindly ask to specify what kind of formal analysis the reviewer would expect.
> >
> >
> > **Limitations with respect to behavioral priors**
> >
> > We state that temporal priors are capable of overcoming the limitations of behavioral prior when attempting to transfer knowledge across a large domain gap. As a matter of fact, when the domain gap decreases, and the downstream task is sampled from the distribution of training tasks, behavioral priors remain superior (Fig. 5, ‘reach’ and ‘u-maze’). Moreover, temporal priors are not able to represent complex interactions with objects as of now. This issue could be solved by investigating state-dependent temporal priors, which we leave as future work. We added a further discussion as Appendix E.
> >
> > **Additional seeds**
> >
> > We have increased the number of seeds for each experiment by 2; results remain consistent with our claims. We plan to run 2 additional seeds by the end of the rebuttal window, for a total of at least 7 seeds per experiment.
> >
> >
> > **How does the policy’s entropy quantify the agent’s confidence to reach the goal?**
> >
> > If multiple actions led to the same outcome, then the policy’s entropy could still be high. This is indeed a legitimate concern: high entropy does not necessarily represent low confidence in reaching the goal. On the other hand, the opposite is in general true: when no reward is observed and the agent could not produce good trajectories, entropy is maximized instead. We empirically observed that entropy remains a reasonable measure within our settings. We added a more detailed description of this issue in Appendix E, which is dedicated to assessing the limitations of our method in more detail.
> >
> > **Results for PARROT in Fig. 5**
> >
> > Figure 5 only shows trajectories for methods that do not need to be conditioned on a goal to act. Adding results for PARROT would require defining a set of goals, which we assume are not available in this experimental section.
> >
> > **Temporal prior as a parameterized action space**
> >
> > The proposed integration of temporal priors does not work in practice. This is because the agent’s actions $a_t$ would be transformed depending on past actions (e.g. $a_{t-1}$), which would break the Markovianity of the environment from the agent’s perspective. In other words, when the agent observes state $s_t$ and chooses action $a_t$, this action would be transformed according to past information.
> >
> > Let us consider a single-step state-independent temporal prior $\bar \pi(a_t|a_{t-1})$. The distribution of the next state $s_{t+1}$ would not only depend on the current state and action: $p(s_{t+1}|s_t, a_t, a_{t-1}) \neq p(s_{t+1}|s_t, a_t)$.
> > Intuitively, the actor is forced to produce an action, but its actions will be transformed in a way that it cannot predict, due to lack of access to the conditioning variables. When tested empirically, this method severely underperforms.
> >
> >
> > **Connection with implicit task inference**
> >
> > We thank the reviewer for this suggestion. While this is not explicitly encouraged, temporal priors trained on several tasks could in principle learn to perform implicit task inference in order to accurately model the distribution of the next action. We have added a brief discussion of this perspective in Appendix G.
> >
> >
> > **References**
> >
> > [1] Florensa, Carlos et al. Stochastic Neural Networks for Hierarchical Reinforcement Learning. ICLR 2017
> >
> > [2] Haarnoja, Tuomas et al. Latent Space Policies for Hierarchical Reinforcement Learning. ICML 2018

---

> > > ### Comment · Reviewer_eq2m · 2021-11-21
> > > **Response follow-up**
> > >
> > > **Additional meta-world tasks**
> > >
> > > There appears to be a gap between the performance in the originally reported experiments and the remaining tasks in MT-10. In 4/5 of the remaining tasks the proposed algorithm either fails to learn anything or appears to be very inconsistent, while in the remaining task all baselines obtain very similar results (except for the authors' implementation of PARROT). To get a better understanding of the effectiveness of TempoRL on this benchmark, it would be useful if the authors could provide the performance curves for the average success rate on the various tasks (since this is the usual metric used for evaluation on meta-world).
> > >
> > > **Effect of optimization objectives (eq. 4-7) on the sampling distribution policy**
> > >
> > > Unlike the proposed algorithm, SAC optimizes its objectives w.r.t. its maximum-entropy stochastic policy used for exploration. When collecting test rollouts, SAC selects actions with a deterministic policy, ignoring the stochastic component. Thus, SAC's authors can provide formal mathematical intuition w.r.t. the exploratory behavior of their algorithm. As TempoRL's main benefit should be improved exploration it would be very useful to include a similar analysis to provide some formal intuition about its exploration properties.

---

> > > > ### Author Response · Authors · 2021-11-22
> > > > **Response follow-up**
> > > >
> > > > We thank the reviewer for the quick reply.
> > > >
> > > > **Additional meta-world tasks**
> > > >
> > > > We believe that the gap in performance of all methods is caused by the increased difficulty of the additional tasks. In particular, the 2 tasks with higher variability between seeds (`button-press` and `drawer-open`) require very precise control of the end-effector, while the 2 tasks in which no method makes progress (`pick-place` and `push`) often involve grasping, which cannot be reproduced by a temporal prior learned on a simple reaching task, as we discuss in Appendix E.
> > > >
> > > > As requested by the reviewer, we have added the plot showing the mean performance over all the MT10-tasks to Figure 9 (excluding `peg-insert-slide` since the reward cannot be extracted for hindsight relabeling, see Appendix E). The results are averaged over 7 seeds and the shaded area represents the standard deviation across different seeds.
> > > >
> > > > **Effect of optimization objectives (eq. 4-7) on the sampling distribution policy**
> > > >
> > > > As the reviewer has correctly observed, unlike SAC, the objective in our method is not optimized w.r.t. to the (mixture) policy used for exploration, but rather w.r.t. the trainable policy $\pi$.
> > > >
> > > > SAC’s authors state that their objective “encourages exploration by increasing the value of regions of state space that lead to high-entropy behavior” [3].
> > > > If the TempoRL objective was optimized w.r.t. the mixture used for collecting data, exploration would target areas in which the mixture has high-entropy. This would not necessarily be desirable in our setting, as we seek correlated and directed behavior.
> > > > This is the reason why the objective is instead optimized w.r.t. the policy $\pi$ alone: we desire to visit areas in which only the policy $\pi$ (not the mixture) is uncertain and the likelihood of sampling from the temporal prior $\bar \pi$ is increased.
> > > >
> > > > As a consequence, the regions of state space explored by our method are those that produce high mixing weights $\lambda$ and not necessarily those in which the sampling policy (mixture) has high entropy.
> > > > This is because, while the entropy of $\pi$ will be high in unexplored areas, the mixing weight $(1-\lambda)$ assigned to it will be small, and the entropy of the sampling policy $(\lambda \bar \pi + (1-\lambda) \pi)$ will be dominated by the temporal prior $\bar \pi$.
> > > >
> > > >
> > > > **References:**
> > > >
> > > > [3] Haarnoja, Tuomas et al. Soft Actor-critic: Off-policy Maximum Entropy Deep Reinforcement Learning with a Stochastic Actor. ICML 2018

---

### Official Review · Reviewer_xn1S · 2021-11-02

**Correctness:** 2
**Technical Novelty And Significance:** 4
**Empirical Novelty And Significance:** 4
**Recommendation:** 8
**Confidence:** 5

**Main Review:**

The idea is novel, simple and shows good empirical results. They use a good choice for the baselines and show that their methods outperforms them on environment from the meta-world suite.

On the weaknesses side, I've found the description of SAC + TempRL quite unclear and not exactly correct:

First, given the objective that takes $\Lambda$ in the formulation (eq 4), the soft-policy target (soft-max of the Q-functions in SAC) may differ, but is not described.

Then, the objective $J_\pi$ looks like the target of the value function $V_\psi$ in SAC, if the expectation was on the action instead of the states. In sac, the objective for the policy $J_\pi$ should be the KL divergence between the policy and the soft-max of the Q-function. Only the formulation of $J_\theta(Q) looks correct.

The parameters $\psi$ and $\theta$ should appear in the objectives function to understand what 'update $\psi$ and $\theta$ means in Algo.1.

And since, at the end, these objectives are not used but they simply used the original SAC, why not just say, after equation 3:
"To directly encourage sampling from the exploration prior, we use an entropy-regularized RL algorithm, for instance SAC" ?

Regarding the experiment, I was expecting more type of environment, to show that this method does not only work on the meta-world suite. For example, using Mujoco or even tasks with discrete actions.



**Summary Of The Paper:**

This paper suggests a task-agnostic exploration priors that can be trained on policies that solve random tasks in a given environment (task-agnostic data). Given a new task, the induced exploration quickly finds how to solve it, compared to other methods that disposes of the same pre-training data.

Temporal priors learns by imitation (of task-agnostic data) a policy conditioned by the whole history, rather than only the current states.
Here, they observe that only using a condition on the previous action is already sufficient to outperforms other methods.

To use the prior, they use an $\epsilon$-greedy approach where $\epsilon$ ($\lambda_t$ in the paper) depends on the entropy of the exploitation policy: the more the exploitation policy is random, the more the agents refers to the exploration policy (the temporal prior).

In order to favorise the utilisation of the exploration, they use SAC which enforces the entropy of the exploitation policy.


**Summary Of The Review:**

+) novel and simple approach and nice results on meta-world.

-) Unclear and useless SAC + TempRL description in section 4.2 and appendix A. Needs more environments for experiments.

I am on the fence on this paper. Since the results are nice and the method is corrects (they finally used the original SAC), I would be more favorable for a weak accept, but I strongly suggest to remove the paragraph about the entropic regularisation in section 4.2 as well as the appendix A and re-write Algo1 to explicite that the used RL algorithm to learn the exploitation policy is SAC.

---

> ### Author Response · Authors · 2021-11-20
> **Response to xn1S**
>
> We thank the reviewer for the comments and constructive feedback. We will now address their concerns.
>
>
> **Description of SAC + TempoRL**
>
> We note that our algorithm actually uses the modified objectives (eq. 5 and 6), because we observe a performance improvement in our evaluation as shown in Appendix. B. In this Appendix we also report that optimizing the unmodified objectives from SAC could remain a viable option for simpler tasks.
>
> To address the concerns on correctness, we have extended Appendix A to make our derivations more clear. In particular, $J_\pi$ is derived from an alternative formulation of SAC. Instead of training the policy by minimizing the KL-divergence between policy and soft-max of the Q-function [1], one can alternatively train it to maximize the value function [2] (without breaking Lemma 2 and derived results in [1]).
>
> To see this, we can take the objective in eq. 10 from [1], adopting the notation used in our paper:
>
> $$
> J_\pi(\theta) = E_{s_t \sim \mathcal{D}} \bigg[ D_{KL}\bigg( \pi_\theta (\cdot |s_t) || \frac{\exp(Q_\phi(s_t, \cdot))}{Z_\phi(s_t)}\bigg)\bigg]
> $$
>
> We can now rewrite the optimization objective as:
>
> $$
> argmin_\theta J_\pi(\theta) = argmin_\theta E_{s_t \sim \mathcal{D}} \bigg[ D_{KL}\bigg( \pi_\theta (\cdot |s_t) || \frac{\exp(Q_\phi(s_t, \cdot))}{Z_\phi(s_t)}\bigg)\bigg]
> $$
> $$
> = argmin_\theta E_{s_t \sim \mathcal{D}, a_t \sim \pi_\theta} \bigg[\log(\pi_\theta (a_t|s_t)) - Q_\phi (s_t, a_t) + \log(Z_\phi(s_t))\bigg]
> $$
> $$
> = argmin_\theta E_{s_t \sim \mathcal{D}, a_t \sim \pi_\theta} \bigg[ \log(\pi_\theta (a_t|s_t)) - Q_\phi (s_t, a_t) \bigg]
> $$
> $$
> = argmax_\theta E_{s_t \sim \mathcal{D}, a_t \sim \pi_\theta}\bigg[ Q_\phi (s_t, a_t) - \log(\pi_\theta (a_t|s_t)) \bigg]
> $$
> $$
> = argmax_\theta \bigg[ V_\phi(s_t) \bigg] ,
> $$
>
> where we used that $Z_\phi(s_t)$ does not depend on parameter $\theta$ and the definition of value given in eq. 3 from [1]:
>
> $$
> V_\phi(s_t) = E_{a_t \sim \pi}[Q_\phi(s_t, a_t) - \log \pi_\theta(a_t|s_t)].
> $$
>
> In the case of SAC, when using the value function defined in eq. 3 from [1], these two formulations result in the same policy objective. However, when using the value definition in [2], eq. 3, a scaling factor $\alpha$ is introduced into the policy objective when directly optimizing the value function.
> In our case, minimizing the KL-divergence between the policy and the soft-max of the Q-function results in
>
> $$
> J_\pi = -\mathop{\mathbb{E}}_{s \sim \mathcal{D}} \bigg[Q_\phi^\pi(s, a) -\log \pi_\theta(a|s) \bigg] \quad \text{with } a \sim \pi_\theta(\cdot|s)
> $$
>
> while maximizing the value leads to
>
> $$
> J_\pi = -\mathop{\mathbb{E}}_{s \sim \mathcal{D}} \bigg[Q_\phi^\pi(s, a) + \alpha \Lambda(-\log \pi_\theta(a|s)) \bigg] \quad \text{with } a \sim \pi_\theta(\cdot|s).
> $$
>
> In practice, we found the first alternative to perform similarly to the simplified method in Appendix B, which only differs for the presence of the scaling factor $\alpha$. For this reason, we have chosen the second alternative. On a side note, as the reviewer has recommended, we have included the parameters in the objective functions for clarity.
>
>
> **Additional environments**
>
> TempoRL was designed for transferring knowledge in sparse, long horizon tasks. This narrows the choice of environments down to those which present these qualities, incorporate multiple tasks and are suitable for collecting expert trajectories. We additionally experimented with continuous control tasks from Mujoco to demonstrate more general applicability of our method. Results are consistent with other suites available in Appendix F.3.
> Moreover, we added results on a more complex maze structure in Appendix F.2 as well as on several more tasks from `meta-world` (see Appendix F.1).
>
>
> **References:**
>
> [1] Haarnoja, Tuomas et al. Soft Actor-critic: Off-policy Maximum Entropy Deep Reinforcement Learning with a Stochastic Actor. ICML 2018
> [2] Haarnoja, Tuomas et al. Soft actor-critic Algorithms and Applications. 2018

---

> > ### Comment · Reviewer_xn1S · 2021-11-22
> > **Satisfying response**
> >
> > I thank the authors for taking the time to better explain the derivation of the policy's objective.
> > I also apologies for misunderstanding the usage of the modified objectives that appear to improve the performance.
> >
> > Regarding the additional environments, I am tempted to say that only adding the swimmer task from Mujoco looks a bit cherry-picked, but is understandable in the limited time for addressing the rebuttal concerns.
> >
> > Beside, I really liked the idea of training an exploration behaviour from a dataset formed by solving random goals, and I believe this one should be communicated and desserves a publication. Therefor I will change my rating for a favorable acceptance.

---

> > > ### Author Response · Authors · 2021-11-23
> > > **Re: Satisfying response**
> > >
> > > We thank the reviewer for the fast reply. We plan to report the results for more complex MuJoCo environments, e.g. `hopper-v2` and `halfcheetah-v2`, for the camera-ready paper if it is accepted for publication. Due to time- and resource-constraints, we have not been able to run them during the rebuttal phase.

---

### Official Review · Reviewer_j8Xm · 2021-11-03

**Correctness:** 4
**Technical Novelty And Significance:** 4
**Empirical Novelty And Significance:** 3
**Recommendation:** 8
**Confidence:** 4

**Main Review:**

The paper is well written and clear. The main contributions are novel, interesting, and well motivated. The experimental results seem informative and in support of the claims made throughout the paper, but the low choice of seeds leaves room for doubt in some cases.

The decision to only use 3 seeds greatly weakens the significance of these results, and is my only major concern with this work. For instance, the low seed count is likely why in the "reach" plot in Figure 4, the 2 and 5 action results appear lower than the 1 and 10 action. Similarly, it is very difficult to conclude anything from the TempoRL-simplified vs TempoRL comparison as a result of the low seed count. With so few seeds, a single bad run can drastically alter the mean. This issue of significance doesn't seem as bad in the case of Figure 6 which use 5 seeds and also shows some consistent patterns across the different domains so I don't think this necessarily invalidates all the empirical results.

Similarly, with so few seeds, e.g., 3 or even 5, the empirical standard deviation can be very noisy which makes it a very poor representation of the expected variance in the results. With so few seeds, plotting the min, max, and median or mean would provide a much more representative visualization of the learning behavior of each method. Alternatively, with few seeds, it isn't unreasonable to simply plot each learning curve with the addition of curve representing the mean/median (see [1, Figure 1] for an example of what I mean).

Overall, I think the main idea is good, and the work is interesting. If it weren't for the aforementioned issues, this would be a strong paper. That being said, given not all results are equally affected, I don't think it fair to reject. I feel like the paper would still be borderline if you were to remove the problematic results which is my recommendation for now.

# Questions:

- p. 6, Why do we need a new objective, i.e., eq. (4)? What is the motivation for wanting policies that are less certain to reach the goal?

- p. 6, "We finally note that the modified learning objective remains aligned with the original formulation.", it's not quite clear to me what aligned means here. Is this a formal statement or an observation based on empirical evidence? If it's the latter, then what is the evidence of their alignment?

- Appendix A, eq. (9) and (10) seem to differ by a missing $\Lambda (\mathcal{H}(\pi(\cdot|s_t)))$ term. What is the motivation for dropping this term?

- Appendix D.1, cumulative returns per episode?

- Appendix D.2, What were the scripted policies that generated data for the priors in "reach" and "room-maze"?


# Minor comments and nitpicks:

- p. 3, "the issue of temporal correlation is merely relocated in the hierarchy, as the high-level planner is not encouraged to produce correlated sequences of skills", I'm not sure this is a fair assessment. Although it is true that the high-level planner is not encouraged to produce correlated sequences of skills, the skills themselves will generate correlated sequences of actions. The issue of temporal correlation isn't relocated so much as implemented at the skills/options level and  abstracted away at the high-level. Doing so doesn't change the fact that when the high-level planner executes just a single skill, it will be producing a correlated sequences of primitive actions, by construction, regardless of any encouragement to do so.

- p. 3, definition of goal-conditioned MDP, the cited work doesn't seem to use the same definition, most notably it doesn't include a discount factor in its definition and the distribution over goals is part of the definition. They don't need to match but I would make it clear in the wording.


[1] Fujimoto S, Meger D, Precup D. Off-policy deep reinforcement learning without exploration. InInternational Conference on Machine Learning 2019 May 24 (pp. 2052-2062). PMLR.

Post-Rebuttal
=========

The increase in number of seeds and the corrections of the error in the one of the figures gives me significantly more faith in the empirical results. I've increased my scores to reflect this.

**Summary Of The Paper:**

The authors propose a novel method for encouraging exploration based on the idea of behavior priors. The proposed prior, called temporal prior, considers the history of the agent when determining the prior probability of an action. The authors then derive an variant of soft-actor critic (SAC) adapted to make use of these temporal priors. This is done by defining a behavior policy that is a mixture of the priors and the agent's learned policy based on the certainty of the agent's policy, as defined by its entropy in that state. Additionally, the authors propose using a custom loss meant to encourage uncertainty in the learned policy and therefore encourage actions sampled from the temporal priors. This approach is shown to encourage state coverage, to be capable of accelerating learning in unseen and different tasks.

**Summary Of The Review:**

The paper is well written and clear. The main contributions are novel, interesting, and well motivated. The experimental results seem informative and in support of the claims made throughout the paper, but the low choice of seeds leaves room for doubt in some cases.

---

> ### Author Response · Authors · 2021-11-20
> **Response to j8Xm (1/2)**
>
> We thank the reviewer for the extensive feedback and valuable suggestions. We will address each one separately.
>
>
> **Number of seeds**
>
> We have increased the number of seeds for each experiment by 2. The updated results remain consistent with claims made in the paper. We plan to run 2 additional seeds by the end of the rebuttal window, bringing the minimum number of seeds to 7 for all experiments. This will further reduce the variance of the empirical standard deviation, which should address the concern about plotting mean/std instead of min/max/median.
>
> As a side remark, we found that Figure 4 contained an error: the colors for $\pi(a_t|s_t)$ and $\pi(a|s_t,a_{t-1})$ were swapped with colors for $\pi(a_t|a_{t-2}^{t-1})$ and $\pi(a_t|a_{t-5}^{t-1})$. In this setting, state-conditional priors underperform as discussed in Appendix D.4, paragraph on state-conditional priors. Temporal priors perform similarly for all conditioning sequence lengths. This has now been fixed.
>
>
> **Why do we need a new objective, i.e., eq. (4)?**
>
> Similarly to SAC, optimizing the objective in eq. (4) maximizes rewards together with an additional regularization term. In SAC this regularization term attempts to maximize $\mathcal{H}(\pi(\cdot|s))$. This results in sampling diverse actions, but fails to produce correlated exploration.
> In our framework, we have access to a temporal prior $\bar \pi$, which is able to produce better exploration trajectories. For this reason, our regularization term should not simply maximize the policy’s entropy, but rather increase the probability of sampling from the temporal prior.
>
> To allow this, we sample actions from a mixture between the policy and prior at training time, and replace the entropy regularization term with a term that encourages large mixing weights, which would result in sampling more often from the prior.
> This mixing weight could be produced by a learned function of the state. However, this would add additional complexity to our method. In practice, we found that it is sufficient to compute the mixing weight as a fixed function $\Lambda(\cdot)$ of the policy’s entropy. As a consequence, the regularization term encourages uncertain policies in order to increase the likelihood of sampling from the prior and improve exploration and state space coverage.
>
> Due to our choice of mixing function, the new regularizer is maximized by the same (uniform) policy that maximizes SAC’s regularization term. However, the complete objectives are not equivalent. An empirical validation of the change of objective is found in Appendix B.
>
>
> **Alignment of modified objective with the original formulation**
>
> p. 6, "We finally note that the modified learning objective remains aligned with the original formulation." is indeed not a formal statement. We have rewritten the first paragraph of Appendix B to address this concern. In short, while the new objective is not necessarily maximized by the maximizer of SAC’s objective, (1) the reward terms $E_{\tau \sim \pi}\bigg[\sum_{t=0}^{\infty}\gamma^t R(s_t, a_t) \bigg]$ are identical and (2) the regularizers are maximized by the same policy due to monotonicity of the mixing function. As these observations might suggest, we empirically observe in Appendix B that optimizing an unmodified objective also improves exploration, although the final performance is lesser than that of the full method. This is the reason why we use the informal concept of alignment.
>
>
> **Appendix A, eq. (9) an (10)**
>
> We decide to not include the term $\Lambda(\mathcal{H}(\pi(\cdot|s_0)))$ in the Q-function to remain consistent with the notation in [1]. We note that this choice remains arbitrary, and different definitions are used in different papers.
>
> We noticed that we omitted the distributions with respect to which the entropy is computed in eq. (10) and eq. (11). This has now been corrected. We thank the reviewer for the pointer.
>
> **Appendix D.1 and cumulative returns per episode**
>
> Yes, cumulative returns are computed per episode. We changed the wording to make this explicit.
>
>
> **Appendix D.2 and scripted policies**
>
> Scripted policies are simply used to collect task-agnostic expert trajectories for training temporal or behavioral priors. Scripted policies are designed to move the agent to randomly sampled goal positions in a space without obstacles (as in `room-maze` and `reach`). Since actions in these environments directly control the position of the agent/end-effector, scripted policies simply receive the current 2D (or 3D) position of the agent and goal, and output the scaled difference between the two. Actions are then corrupted with Gaussian noise. We finally note that scripting a policy is only one way to generate expert data, which can also be collected via human supervision or by an RL agent trained from scratch. We added more details on this in Appendix. D2. A full implementation of scripted policies is available as part of the submitted code.

---

> > ### Author Response · Authors · 2021-11-20
> > **Response to j8Xm (2/2)**
> >
> > **Temporal correlation in hierarchical RL**
> >
> > We agree with the reviewer’s assessment. Skills themselves are usually capable of producing correlated sequences of actions. However, we believe that abstracting away temporal correlation at the highest level of a hierarchical framework might not be effective in practice. Considering a task that requires the execution of multiple skills in a precise order: if skills of fixed lengths are used to learn this task, the duration of the required correlated behavior might be longer than the duration of a single skill. For this reason, the highest layer could also benefit from temporally correlated behavior.
> >
> > For instance, using action repeat can be seen as a high level policy acting on fixed skills. In this case, temporally correlated behavior is not encouraged to propagate for longer than the duration of a single skill. When using temporal priors, the action taken at each time step directly depends on the previous action. This is not the case for HRL approaches at timesteps in which a skill terminates and a new one is selected.
> >
> >
> > **Definition of goal-conditioned MDP**
> >
> > Thanks for pointing this out. We clarified that we deviate from the original definition.
> >
> >
> > **References**:
> >
> > [1] Haarnoja, T. et al. Soft Actor-Critic Algorithms and Applications, ICML 2018

---

> > > ### Comment · Reviewer_j8Xm · 2021-11-22
> > > **Response**
> > >
> > > The improvements made to the paper address my major concerns. I'll be increasing my score to reflect this.

---

> > > ### Comment · Reviewer_j8Xm · 2021-11-23
> > > **Number of seeds**
> > >
> > > The current draft only mentions 3 seeds but the rebuttal mentions additional seeds. Can the authors confirm that the number of seeds in the results has been updated?

---

> > > > ### Author Response · Authors · 2021-11-23
> > > > **Re: Number of seeds**
> > > >
> > > > We thank the reviewer for the discussion and for re-evaluating our submission.
> > > >
> > > > We can confirm that the plots in the paper revision contained the updated experiments (at least 5 seeds), and apologize for not adjusting the text accordingly. Importantly, as promised in the answer above, we have now finalized adding another 2 seeds to the latest revision. As a consequence, all experiments are now run over 7 seeds (and 9 seeds for Figure 6), and the text in the Appendix has been updated accordingly. We apologize again for the inaccuracy in the previous draft and thank the reviewer for the pointer.

---

### Official Review · Reviewer_AE8g · 2021-11-26

**Correctness:** 2
**Technical Novelty And Significance:** 2
**Empirical Novelty And Significance:** 2
**Recommendation:** 3
**Confidence:** 3

**Main Review:**

The general premise of the paper is that learning priors on actions conditioned on past actions allow the agent to have a reasonable directed behaviour even when the observation space of the data used to learn the priors is different from the observation space of the RL problem.

For the general RL problem, the premise of the paper is false --- one can design an offline dataset, or design a downstream task such that priors on actions conditioned on past actions from an offline dataset would hurt more than help. Similarly, one can design a problem for which priors conditioned on the state would outperform priors conditioned on just a sequence of actions.

For instance, imagine a grid-world in which the policy for the offline dataset is to go right with 97% probability, and the other directions with 1% probability, whereas the optimal policy on the downstream environment is to always go up. Action priors conditioned on past actions would only mislead the agent in such an enviroment. Similarly, an environment in which different parts of state space require very different action sequences for directed exploration would clearly benefit from priors conditioned on the state instead of past actions.


As a result, I find the primary claim "However, an agent should ideally be able to produce efficient explorative behaviors even in unseen environments and unrelated tasks. Behavior priors don’t give us that, temporal priors must." unfounded. The priors can absolutely hurt when the downstream task is unrelated to the offline data, and a few empirical results are not sufficient to support the broad claim. The fact that a prior over actions only conditioned on the last action works only tells me that achieving directed exploration in the benchmarks the authors considered is not particularly challenging, and a simple prior enables directed exploration across the state space.

**Summary Of The Paper:**

The paper proposes a method to incorporate prior knowledge in an RL system to improve exploration. The authors propose learning a prior over actions conditioned on either the state, or just past actions and show that priors conditioned on just past actions generalize better than those conditioned on the full state when the observation space of offline data is different from the RL problem.


**Summary Of The Review:**

The main claim of the paper is not supported by the evidence. It's not hard to imagine scenarios for which the main claim can be demonstratively false. In my opinion, the paper must clearly mention when the proposed method would help and hurt, and tone down the main claim.

---

> ### Author Response · Authors · 2021-11-27
> **Response to AE8g**
>
> We thank the reviewer for the assessment.
>
> **Usefulness of Priors**
>
> We would like to note that the concerns raised apply to the entire body of work spanning behavioral priors, offline RL and most methods which tackle learning from demonstrations: the quality of the learned prior/policy depends on the quality of the data. It is generally known that an uninformative or even adversarial prior, as the one outlined in the review, can harm the performance of methods that make use of such a prior. Yet, many methods successfully leverage offline data and demonstrations  [1,2,3].
> Since we operate in the context of these works, assuming that previous methods are not invalid, we argue that our submission does constitute a significant contribution over the existing state-of-the-art in the proposed settings (large gap in state space but small gap in action distribution), as we empirically show in Section 5.
>
> In alignment with previous works, temporal priors are data-driven and, as such, their effectiveness heavily relies on the trajectories they are trained on (please see Appendix E for a thorough discussion on these limitations). We do not claim that our method is immune to adversarially designed datasets or tasks.
> In practice, we found that, when training trajectories are generated by completing simple tasks (such as reaching random goals), temporal priors can successfully generalize to more complex, unseen tasks, e.g. the `window-open` task (Section 5.2 and Appendix F).
> As reported in previous works [1], we also observe that behavioral priors perform well when trajectories in the dataset were collected on a task that is very similar to the downstream task (Section 5.2). However, we found that behavioral priors struggle when presented with observations that are out-of-distribution with respect to the training data, as is the case when deploying them across a domain gap and in unseen tasks (e.g., when having an unseen object in the scene in the downstream task). In this case, we suggest that action sequences in the dataset may still be informative and suitable to reconstruct correlated behavior.
>
> We can also more formally address the settings in which temporal and behavioral priors would help or fail. Let us define the joint state-action distribution in the offline dataset as $p^{\mathcal{D}}(s_t, a_t)$ and the distribution of consecutive actions as $q^{\mathcal{D}}(a_{t-1}, a_t)$. Similarly, let us now introduce the same distributions for an (unknown) optimal policy in downstream learning as $p^{\pi^\star}(s_t, a_t)$ and $q^{\pi^\star}(a_{t-1}, a_t)$. Behavioral priors in principle require a small distance measure between $p^{\mathcal{D}}$ and $p^{\pi^\star}$, while state-independent temporal priors on the other hand rely on similarly distributed $q^{\mathcal{D}}$ and $q^{\pi^\star}$.  We are happy to integrate this statement into the paper in order to clarify the scope of our contribution with respect to generalization capabilities of prior-based RL methods.
>
> **Additional Comments**
>
> We back our claims with an empirical analysis, which was significantly extended during the rebuttal phase to include all available tasks from the MT10 benchmark, three `point-maze` structures and a proof of concept in a new setting (Appendix F.3, where we show that the method helps when dealing with noisy observations in the downstream task). Across these environments, we found that achieving directed exploration without hand-designed strategies is not straightforward (see Appendix F.6), and to the best of our knowledge no method can effectively leverage task-agnostic trajectories to achieve this.
>
> In order to address the raised concerns, we will tone down our claims in accordance with the suggestions, particularly for the statement the reviewer quoted.
>
> **References:**
>
> [1] Singh, Avi et al. PARROT: Data-driven Behavioral Priors for Reinforcement Learning. ICLR 2021
>
> [2] Kumar, Aviral et al. Conservative Q-Learning for Offline Reinforcement Learning. NeurIPS 2020
>
> [3] Rajeswaran, Aravind et al. Learning Complex Dexterous Manipulation with Deep Reinforcement Learning and Demonstrations. RSS 2018

---

### Author Response · Authors · 2021-11-23
**General Response**

We thank all the reviewers for their extensive feedback and comments. They have helped to greatly improve the paper during the rebuttal phase. Given the large number of reviews, we have decided to provide a short overview of the main changes and additions to the paper:

- The number of seeds for all experiments was increased from 3 (5 for Figure 6) to 7 (9 for Figure 6). The results are consistent with those originally reported.
- The experimental validation was extended to include 5 additional meta-world tasks, and hence contains all suitable MT10 tasks.
- We have added an additional, more complex, maze structure.
- To show that the method can in principle also work in domains other than the presented `meta-world` and `point-maze` suite, we provide a proof-of-concept experiment on a MuJoCo task (`swimmer-v2`) (Appendix F).
- A thorough discussion of limitations was introduced as Appendix E.
- An ablation for HER and n-step returns and a study on action repeat were added to Appendix F.
- The clarity and notation of our method (Section 5 and Appendix A, B, C) were improved.
- The related works section was extended as Appendix G.

---

### Decision · Program_Chairs · 2022-01-20

**Decision:**

Reject

**Comment:**

This paper was close and also very polarizing with the reviewers. On the positive side, some reviewers found:
1. the results impressive
2. the proposed method to be novel, interesting, and produce good performance across several settings
3. the paper was well written

On the other hand, others found:
1. the motivation suspect
2. missing experiments to characterize the sensitivity to numerous hyper-parameters
3. the baselines compared with weak and not representative
4. significant performance drop comparing the results in the original submission and the new ones added during discussion period
5. low number of seeds initially

In the end, multiple reviewers raised serious issues regarding the motivation for the approach and the quality and ultimately credibility of the results presented. One of the high-scoring reviewers agreed the paper was a bit misleading (limitations relegated to the appendix). Unfortunately, none of the high-scoring reviewers provided counters to this points.